# Live imaging of the co-translational recruitment of XBP1 mRNA to the ER and its processing by diffuse, non-polarized IRE1α

Silvia Gómez-Puerta[1], Roberto Ferrero[1], Tobias Hochstoeger[2,3], Ivan Zubiri[1], Jeffrey Chao[2], Tomás Aragón[1]*, Franka Voigt[2]*

[1]Department of Gene Therapy and Regulation of Gene Expression, Center for Applied Medical Research (CIMA), University of Navarra, Pamplona, Spain; [2]Friedrich Miescher Institute for Biomedical Research, Basel, Switzerland; [3]University of Basel, Basel, Switzerland

**Abstract** Endoplasmic reticulum (ER) to nucleus homeostatic signaling, known as the unfolded protein response (UPR), relies on the non-canonical splicing of XBP1 mRNA. The molecular switch that initiates splicing is the oligomerization of the ER stress sensor and UPR endonuclease IRE1α (inositol-requiring enzyme 1 alpha). While IRE1α can form large clusters that have been proposed to function as XBP1 processing centers on the ER, the actual oligomeric state of active IRE1α complexes as well as the targeting mechanism that recruits XBP1 to IRE1α oligomers remains unknown. Here, we have developed a single-molecule imaging approach to monitor the recruitment of individual XBP1 transcripts to the ER surface. Using this methodology, we confirmed that stable ER association of unspliced XBP1 mRNA is established through HR2 (hydrophobic region 2)-dependent targeting and relies on active translation. In addition, we show that IRE1α-catalyzed splicing mobilizes XBP1 mRNA from the ER membrane in response to ER stress. Surprisingly, we find that XBP1 transcripts are not recruited into large IRE1α clusters, which are only observed upon overexpression of fluorescently tagged IRE1α during ER stress. Our findings support a model where ribosome-engaged, immobilized XBP1 mRNA is processed by small IRE1α assemblies that could be dynamically recruited for processing of mRNA transcripts on the ER.

*For correspondence: taragon@unav.es (TA); franka.voigt@fmi.ch (FV)

Competing interest: The authors declare that no competing interests exist.

## Editor's evaluation

We agree that this study, especially when considered in parallel with the work from Belyy et al., significantly furthers our understanding of how early events in the unfolded protein response pathway trigger downstream signals. This pathway is essential to respond and protect against potentially toxic insults to ER homeostasis. On a more general note, the advances in single-molecule optical imaging, which were developed for your work, will benefit others who wish to probe dynamic signaling events at the ER membrane and beyond.

## Introduction

Cellular organization depends on the ability of cells to recruit mRNA and protein molecules to precise subcellular localizations. In eukaryotic cells, mRNA transcripts that encode membrane and secreted proteins are targeted to the endoplasmic reticulum (ER) to facilitate the efficient and often co-translational delivery of their protein products to the ER lumen. mRNA targeting is mediated through the

co-translational recognition of an N-terminal signal sequence by the signal-recognition particle (SRP; *Walter et al., 1981*). SRP-ribosome-nascent chain complexes are recruited to the surface of the ER by the SRP receptor (*Gilmore et al., 1982*), which channels the nascent polypeptide into the ER lumen through interaction with the Sec61 translocon (*Görlich et al., 1992*).

The unfolded protein response (UPR) acts as a combination of quality control pathways that monitor the folding status of proteins within the ER lumen and adjust the capacity of the ER's folding machinery (*Walter and Ron, 2011*). IRE1α (inositol-requiring enzyme 1 alpha) triggers the most conserved branch of the UPR (*Cox et al., 1993*; *Mori et al., 1993*). It is an ER membrane resident stress sensor that is activated by the accumulation of misfolded proteins in the ER lumen and signals ER stress through the non-canonical splicing of X-box binding protein 1 mRNA (XBP1, *HAC1* in yeast; *Sidrauski and Walter, 1997*; *Tirasophon et al., 1998*; *Yoshida et al., 2001*).

Processing of unspliced XBP1 (XBP1u) mRNA is initiated upon oligomerization and trans-autophosphorylation of IRE1α (*Ali et al., 2011*), which leads to the allosteric activation of its cytosolic kinase and RNAse domains (*Korennykh et al., 2009*). Once activated, IRE1α excises a highly conserved 26 nucleotide intron from the XBP1 coding sequence (*Calfon et al., 2002*; *Yoshida et al., 2001*) and the severed exons are rejoined by the tRNA ligase RtcB (*Jurkin et al., 2014*; *Kosmaczewski et al., 2014*; *Lu et al., 2014*). Intron excision causes a translational frameshift in the spliced XBP1 (XBP1s) transcript, which encodes a potent transcription factor that increases the folding capacity of the ER through a broad activation of stress response genes (*Acosta-Alvear et al., 2007*), including expression of ER-associated degradation factors (*Brodsky, 2012*). Beyond processing XBP1 mRNA, metazoan IRE1α is able to cleave a variety of mRNAs to initiate their rapid degradation in a pathway known as regulated IRE1-dependent decay (RIDD; *Hollien et al., 2009*; *Hollien and Weissman, 2006*). Even though RIDD has been found to play a key role in some pathological conditions, XBP1 splicing stands out as the main physiological output of IRE1 activation (*Ishikawa et al., 2017*).

To efficiently support rapid responses to ER stress, eukaryotic organisms display different strategies to ensure the timely encounter of IRE1α and its substrate mRNAs. In *Saccharomyces cerevisiae*, acute ER stress triggers the rapid oligomerization of IRE1 protein into a discrete number of foci (*Aragón et al., 2009*; *Kimata et al., 2007*). *HAC1* mRNA, the yeast homolog of XBP1, is then recruited into these foci through a bipartite element that is located in the *HAC1* 3' untranslated region (UTR) while translational repression is imposed by the *HAC1* intron itself (*Aragón et al., 2009*; *Rüegsegger et al., 2001*; *van Anken et al., 2014*). This swift targeting of *HAC1* mRNA to pre-formed IRE1p clusters is essential to allow a timely response to ER stress and to sustain yeast proteostasis (*Pincus et al., 2010*).

The activation of metazoan IRE1α has been proposed to follow the same principles that were defined in yeast. Under ER stress, ectopic, fluorescently labeled IRE1α was found to cluster into large dynamic foci, and the kinetics of cluster assembly and disassembly approximately correlated with XBP1 splicing rates (*Li et al., 2010*). Yet, there is no direct evidence that the formation of large IRE1α clusters is required for splicing. Even though oligomerization of IRE1α has been proven to be the regulatory step that coordinates mRNA cleavage (*Korennykh et al., 2009*; *Li et al., 2010*) and the disruption of oligomerization interfaces has been shown to diminish RNAse activity (*Karagöz et al., 2017*; *Sanches et al., 2014*), the specific oligomeric state of splicing-competent IRE1α assemblies has not been precisely determined. In addition, only a minor fraction (~5%) of all cellular IRE1α protein concentrates in detectable foci (*Belyy et al., 2020*) and there is no direct evidence that they are indeed the sites of XBP1 processing at the ER.

In contrast to yeast *HAC1*, metazoan XBP1 mRNA is recruited to the ER surface through co-translational targeting that involves a peptide signal sequence and not a cis-acting localization element. Specifically, XBP1u transcripts encode a hydrophobic stretch (HR2) located at the C-terminal half of the protein that mimics a secretion signal (*Yanagitani et al., 2009*). On translation, this hydrophobic stretch is recognized by SRP, which delivers the nascent chain complex to the Sec61 translocon in the ER membrane (*Plumb et al., 2015*). Recognition of the HR2 peptide is aided by a translational pausing mechanism that has been proposed to stall the translating ribosome through high-affinity interactions with the peptide exit tunnel. This conveys stability to the mRNA-ribosome-nascent chain complex that facilitates its delivery to the ER membrane (*Kanda et al., 2016*; *Yanagitani et al., 2011*). Such a co-translational targeting mechanism suggests that IRE1α encounters XBP1u mRNA at the Sec61 translocon, where translating ribosomes would be poised. This notion is supported by the reported interaction of IRE1α with the translocon complex as well as by crosslinking data that find IRE1α in close

contact with SRP, ribosomal RNAs (rRNAs), and a subset of ER-targeted mRNAs (*Acosta-Alvear et al., 2018*; *Plumb et al., 2015*). However, this model is difficult to reconcile with a situation where IRE1α molecules are recruited into large clusters with complex topologies that are not simply 2D patches in the ER membrane but have also been described to exclude the Sec61 translocon from specific regions within the clusters (*Belyy et al., 2020*).

Here, we have developed a single-molecule imaging approach that visualizes individual XBP1 mRNA transcripts and thus provides an important framework for the investigation of fundamental UPR biology principles and the recruitment of XBP1 mRNA to IRE1α and the ER surface. We image individual XBP1 transcripts that have been recruited for splicing on the ER and show that their recruitment is mediated by a translation-dependent targeting mechanism that involves SRP but functions through a non-canonical signal sequence. We demonstrate that XBP1 mRNAs are mobilized from the ER surface upon induction of ER stress by IRE1α-catalyzed splicing. Using a dual-color live imaging approach, we visualize individual XBP1 mRNA transcripts together with IRE1α-GFP (green fluorescent protein), which only assembles into clusters at increased expression levels and does not stably associate with XBP1 mRNA under splicing inhibition conditions. Instead, when expressed at endogenous levels, IRE1α-GFP simply outlines the ER and cleaves XBP1 mRNA in the absence of cluster formation during ER stress. This finding is further confirmed by a complimentary study that images single IRE1α molecules to characterize their oligomerization dynamics in response to ER stress and also demonstrates that large IRE1α clusters are not required for splicing activity (*Belyy et al., 2021*).

## Results

In order to directly visualize the recruitment of XBP1 mRNA to the ER, we developed a single-molecule imaging approach that takes advantage of the MS2 labeling system to detect individual reporter mRNAs in living cells (*Bertrand et al., 1998*). We generated an XBP1 wild-type (WT) reporter transcript that comprises the complete *Mus musculus* open reading frame (ORF) as well as its complete 3'UTR (*Figure 1A*, red; *Calfon et al., 2002*; *Sugimoto et al., 2015*). To enable the detection of single mRNA molecules at high signal-to-noise ratios, we further included 24 MS2 stem-loops in the 3'UTR of all reporter transcripts (*Figure 1A*) and made use of their specific recognition by fluorescently labeled synonymous tandem MS2 coat proteins (stdMCPs; *Bertrand et al., 1998*; *Wu et al., 2015*).

To complement the XBP1 WT reporter, we introduced a frameshift mutation downstream of the ER intron (*Figure 1A*, yellow, HR2 mutant) to prevent synthesis of the HR2 peptide, which has been shown to be essential for non-canonical SRP-mediated translocation of XBP1u mRNA to the ER membrane (*Kanda et al., 2016*; *Yanagitani et al., 2009*; *Yanagitani et al., 2011*). In addition, we employed a previously characterized SRP-recruited reporter (*Voigt et al., 2017*) to benchmark ER association of XBP1 transcripts against this established reporter construct encoding a secreted Gaussia luciferase protein (*Figure 1A*, gray).

We next generated HeLa cell lines stably expressing these reporter transcripts under a doxycycline-inducible promoter and from single genomic loci (*Weidenfeld et al., 2009*). To allow detection of individual mRNA particles as diffraction limited spots in the cytoplasm of living cells, we co-expressed nuclear localization signal-encoding fluorescently labeled NLS-stdMCP-stdHalo fusion proteins, which recruit excess stdMCP to the nucleus and thereby increase the signal-over-noise ratio in the cytoplasm (*Voigt et al., 2017*).

To confirm that these reporter constructs were indeed splicing competent, we first performed qPCR-based splicing assays (*Figure 1B*). As expected, we detected an increase in the levels of spliced XBP1 WT mRNA (red), and a transient drop in the levels of unspliced WT mRNA in response to the induction of ER stress with thapsigargin (TG). Using these measurements, we calculated the splicing ratio (spliced/unspliced) as a quantitative readout of splicing efficiency. As expected, splicing ratios increased sharply upon induction of ER stress in WT reporter-expressing cells and were much lower in HR2 mutant-expressing cells (yellow). In agreement with the RNA analysis, we detected increased XBP1s protein levels in response to TG treatment in cells expressing WT reporter transcripts (black triangle, *Figure 1C*). HR2-mutant cells produced only residual levels of XBP1s in response to TG treatment (white triangle, *Figure 1C*) while the majority of their XBP1 protein products was still derived from unspliced HR2 mutant transcripts (black triangle, same size as WT XBP1s protein).

We validated that MS2 tagging of ectopic XBP1 mRNA did not compromise its capacity to undergo splicing when ER stress was induced with TG or tunicamycin (TM) and ensured it did not compromise

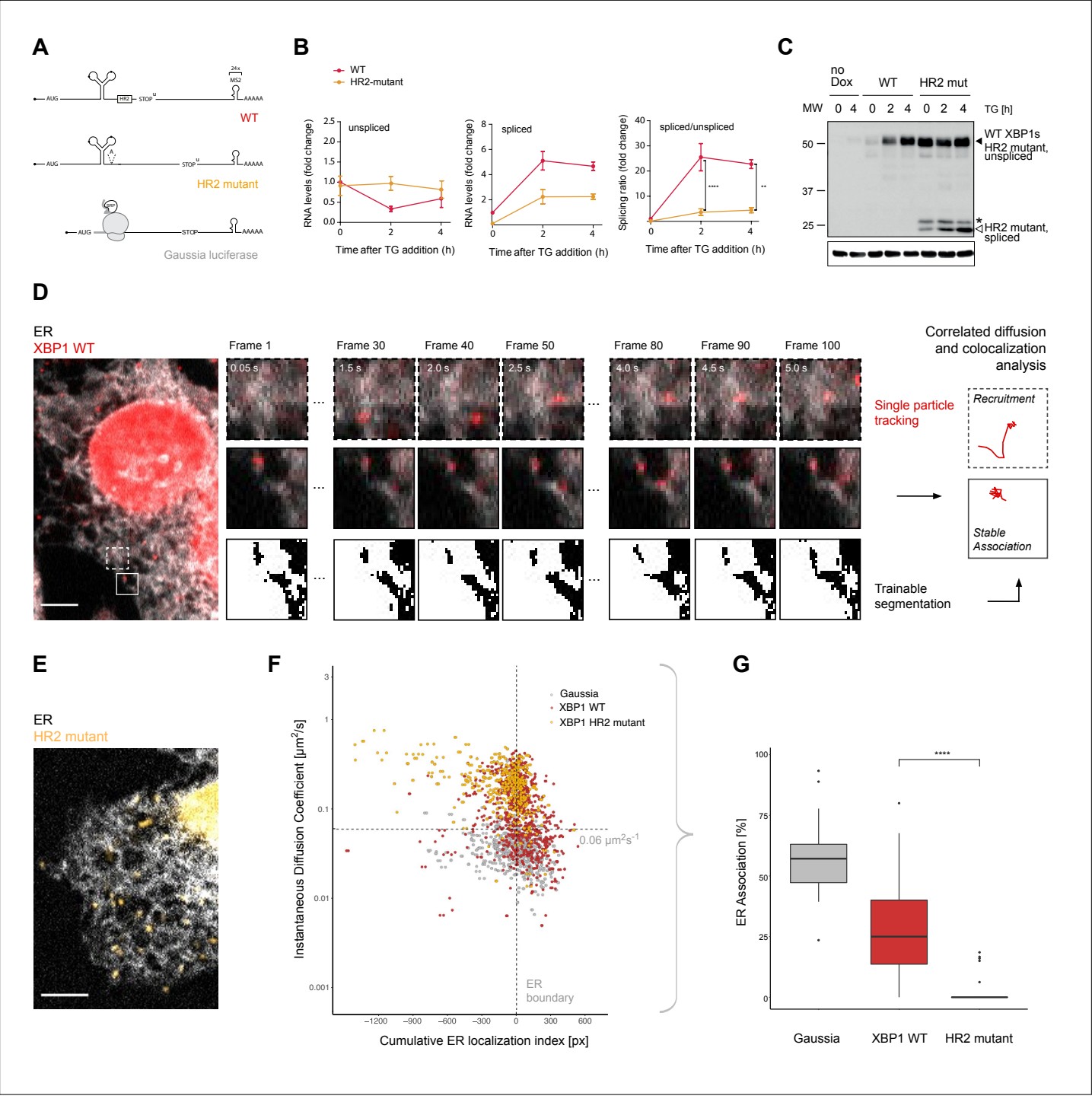

**Figure 1.** Live imaging of XBP1 mRNA recruitment to the endoplasmic reticulum (ER). (**A**) Reporter construct design: XBP1 wild-type (WT; red) features the mouse XBP1 opening reading frame (ORF) and 3' untranslated region (UTR) and contains a 24 × MS2 stem loop array for mRNA detection. XBP1 HR2 mutant (yellow) is identical to the WT construct but contains a point mutation downstream of the ER intron that renders the HR2 peptide out-of-frame. The Gaussia luciferase reporter (gray) is a canonical signal-recognition particle (SRP)-recruited transcript and serves as positive control for ER association. (**B**) qPCR (quantitative polymerase chain reaction) assay showing splicing of MS2-labeled XBP1 reporter transcripts upon induction of ER stress with thapsigargin (TG). HeLa cells expressing WT and HR2 mutant reporters were treated with 0.2 μg/ml doxycycline (Dox) for 15 hours before addition of 100 nM TG for indicated times. Graph indicates the average ± SD (n=3). Statistical test Kruskal-Wallis and Dunn's multiple comparison test. * p<0.05, ** p<0.01, *** p<0.0001 (**C**) Western blot against XBP1 protein in response to unfolded protein response (UPR) activation with 100 nM TG for indicated times using an antibody that does not distinguish between XBP1u/s proteins but preferentially recognizes mouse over human XBP1 (human XBP1s background signal is detectable in samples w/o reporter expression = no Dox). Black triangle: 55 kDa band corresponding to endogenous and

*Figure 1 continued on next page*

*Figure 1 continued*

reporter WT XBP1s, which have the same size as unspliced HR2 mutant protein. White triangle: spliced HR2 mutant XBP1s protein. Asterisk (*): short protein product present before TG treatment. Loading control: Gapdh. (**D**) Representative live-cell image of the XBP1 WT reporter (red) in a HeLa cell expressing NLS-stdMCP-stdHalo and a fluorescent ER marker (gray). Illustration of the image analysis workflow: diffraction-limited spots (*) are individual mRNA transcripts. (**E**) Same as in (**D**) but expressing XBP1 HR2 mutant reporters (yellow). All scale bars = 5 µm. (**F**) Correlated diffusion and ER colocalization analysis of individual XBP1 WT (red), HR2 mutant (yellow), and Gaussia (gray) transcripts. Dots are single particles that were tracked for at least 30 frames. Y-axis: instantaneous diffusion coefficients. X-axis: cumulative ER localization index. Positive values indicate ER colocalization. (**G**) Boxplot showing ER association quantified from data shown in (**F**). Statistical test: unpaired t-test, p-value = 1e-8. For raw data see *Figure 1—source data 1*.

The online version of this article includes the following source data and figure supplement(s) for figure 1:

**Source data 1.** Raw gel images for *Figure 1*.

**Figure supplement 1.** Characterization of XBP1 splicing and unfolded protein response (UPR) activation in XBP1 wild-type (WT) reporter expressing cells.

**Figure supplement 1—source data 1.** Raw gel images for *Figure 1—figure supplement 1*.

**Figure supplement 2.** Flotation assays to investigate HR2-mediated recruitment of XBP1 reporter transcripts to endoplasmic reticulum (ER) membranes.

**Figure supplement 2—source data 1.** Raw gel images for *Figure 1—figure supplement 2*.

the activation of other UPR signaling mechanisms, such as the one initiated by PERK (*Figure 1—figure supplement 1*).

To assess XBP1 mRNA mobility and investigate particle dynamics of individual transcripts, we acquired streaming movies at fast frame rates (20 Hz) that detected XBP1 mRNAs as Halo-labeled diffraction limited spots in the cytoplasm of individual HeLa cells (*Figure 1D*, red). We performed single-particle tracking (SPT) over 100 consecutive frames and used the resulting particle coordinates to determine instantaneous diffusion coefficients (IDCs) as a measure of particle mobility (*Berg, 1993*; *Voigt et al., 2017*).

According to current models, XBP1u WT mRNA (but not the HR2 mutant) should be constitutively recruited to the ER surface for IRE1α-mediated splicing during ER stress. To investigate XBP1 mRNA association with the ER, we therefore integrated a fluorescently labeled ER marker protein (Sec61b-SNAP) into the reporter cell lines introduced above (analogous to *Belyy et al., 2020*). We imaged dual-labeled cells using a fluorescence microscope equipped with two parallel light paths and registered cameras for simultaneous detection of mRNA and ER signal in independent channels (*Video 1*).

Next, we quantified the mobility of individual particles with respect to their ER localization, which not only allowed us to visualize the recruitment of individual mRNAs to the ER surface (*Figure 1D*, upper panels; *Video 2*) but also enabled us to assess whether a particle is stably associated with the ER (*Figure 1D*, middle panels; *Video 3*). To this aim, we segmented the ER signal (*Figure 1D*, lower panels) and used it to generate distance maps that allowed us to correlate particle coordinates with ER boundaries. In these distance maps, positions on the ER were given positive values, and positions away from the ER were defined as negative. Based on particle trajectories, we determined the localization of individual transcripts throughout the entire image series and calculated cumulative ER localization indices that highlight robust localization phenotypes (*Voigt et al., 2017*). We combined the diffusion and ER colocalization analysis and employed it to benchmark the behavior of a Gaussia luciferase reporter transcript (*Figure 1A*)

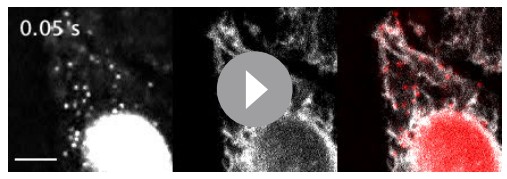

**Video 1.** XBP1 wild-type (WT) mRNA colocalization with the endoplasmic reticulum (ER). HeLa cell line stably expressing XBP1 WT reporter transcripts, NLS-stdMCP-stdHalo, and an ER marker. Simultaneous image acquisition for both channels (XBP1 WT, red, and ER, gray) using 50 ms exposure times (100 frames total). The movie is played at 20 fps. The scale bar is 5 µm.
https://elifesciences.org/articles/75580/figures#video1

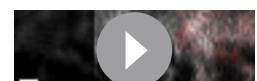

**Video 2.** Recruitment of a single XBP1 wild-type mRNA transcript to the endoplasmic reticulum (ER). Close-up from the same image series as shown in *Video 1* but highlighting an example for a single particle that is recruited to the ER surface.
https://elifesciences.org/articles/75580/figures#video2

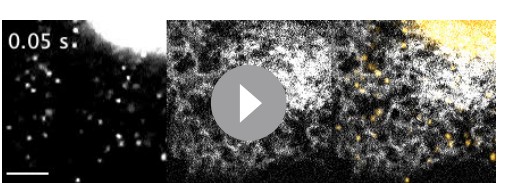

**Video 3.** Stable association of a single XBP1 wild-type mRNA transcript with an endoplasmic reticulum (ER) sheet. Close-up from the same image series as shown in *Video 1* but highlighting an example for a single particle that is stably associated with the ER surface. https://elifesciences.org/articles/75580/figures#video3

that encodes a secreted protein product and that we have previously shown to be predominantly localized to the ER (*Voigt et al., 2017*). Next, we performed the same analysis to quantify the mobility and ER association of XBP1 WT and HR2 mutant reporters (*Figure 1E*).

The combined data show that a large fraction of XBP1 WT transcripts (*Figure 1F*, red dots) behaves similar to the secreted Gaussia mRNAs (*Figure 1F*, gray dots). Many XBP1 WT transcripts exhibit low mobility and colocalize with the ER.

However, there is another population of WT reporter tracks not observed for the Gaussia reporter that is more mobile and tends to not localize to the ER. Interestingly, the behavior of this population is exactly matched by the XBP1 HR2 mutant tracks (*Figure 1F*, yellow dots). These reporter mRNAs seem to have lost their ability to be recruited to the ER surface and exhibit a generally higher degree of mobility that is also apparent upon visual inspection (*Figure 1F*, *Video 4*). We employed the correlated diffusion and ER colocalization analysis to quantify the fraction of ER-associated particles per cell. To this aim, we used the clearly ER-associated Gaussia cluster to define cut-offs (D<0.06 μm²s⁻¹ and positive ER localization index, dashed lines in *Figure 1F*) for identification of XBP1 mRNA particles that showed a similar behavior. Based on these parameters, we found an average (per cell) of 27.4 ± 19.4% (mean ± SD) of all XBP1 WT and 3.1 ± 6.2% of all HR2 mutant transcripts to be associated with the ER (*Figure 1G*).

To corroborate the findings from the single-particle imaging approach through an independent method, we performed membrane flotation assays that allow separation of membrane from cytosolic fractions (*Figure 1—figure supplement 2A*; *Mechler and Rabbitts, 1981*). As expected, we found that XBP1 WT reporter mRNAs were detected in the membrane fractions to a similar extent as the endogenous XBP1u mRNA. In contrast, XBP1u HR2 mutant mRNA lacked membrane association and behaved like endogenous XBP1s (*Figure 1—figure supplement 2B*). Upon reconstitution of the original ORF through integration of two additional nucleotides that restore the HR2 reading frame but not the upstream part of the ORF, membrane association was restored (*Figure 1—figure supplement 2C, D*). Association of XBP1 reporter transcripts with the ER is therefore unambiguously linked to the expression of the HR2 peptide.

Together, these findings indicate that XBP1 WT reporter mRNA is recruited to the ER surface albeit to a lesser extent than canonical secretion-signal encoding Gaussia transcripts. Recruitment depends on the expression of the HR2 peptide, since a reporter mRNA that does not produce HR2 failed to associate with the ER. Our results are consistent with the non-canonical mechanism of XBP1 delivery to the ER and confirm that HR2 expression conveys stable ER association in a co-translational manner.

To test if translation-dependent recruitment of XBP1 transcripts to the ER membrane is necessary to enable mRNA splicing, we generated an XBP1 translation site reporter that would allow us to directly monitor XBP1u translation on the ER (*Figure 2A*). Specifically, we used a nascent polypeptide imaging approach that relies on the co-expression of a well-folded protein scaffold (spaghetti monster, SM) containing nine GCN4 antigen repeats (*Eichenberger et al., 2018*; *Morisaki et al., 2016*; *Yan et al., 2016*) and the MS2 stem loop array introduced above. To quantify protein synthesis of XBP1u transcripts on the ER, we generated a XBP1u translation reporter construct that contains the GCN4-SM downstream of the UPR intron but in frame with the XBP1u ORF. Upon splicing and excision of the intron by IRE1α, the ORF changes to XBP1s and the GCN4-SM is no longer in frame. Thus, the translation site signal can only be detected prior to mRNA splicing.

**Video 4.** Lack of colocalization with the endoplasmic reticulum (ER) exhibited by XBP1 HR2 mutant transcripts. HeLa cell line stably expressing XBP1 HR2 mutant reporter transcripts, NLS-stdMCP-stdHalo and an ER marker. Simultaneous image acquisition for both channels (HR2 mutant, yellow and ER, gray) using 50 ms exposure times (100 frames total). The movie is played at 20 fps. The scale bar is 5 μm. https://elifesciences.org/articles/75580/figures#video4

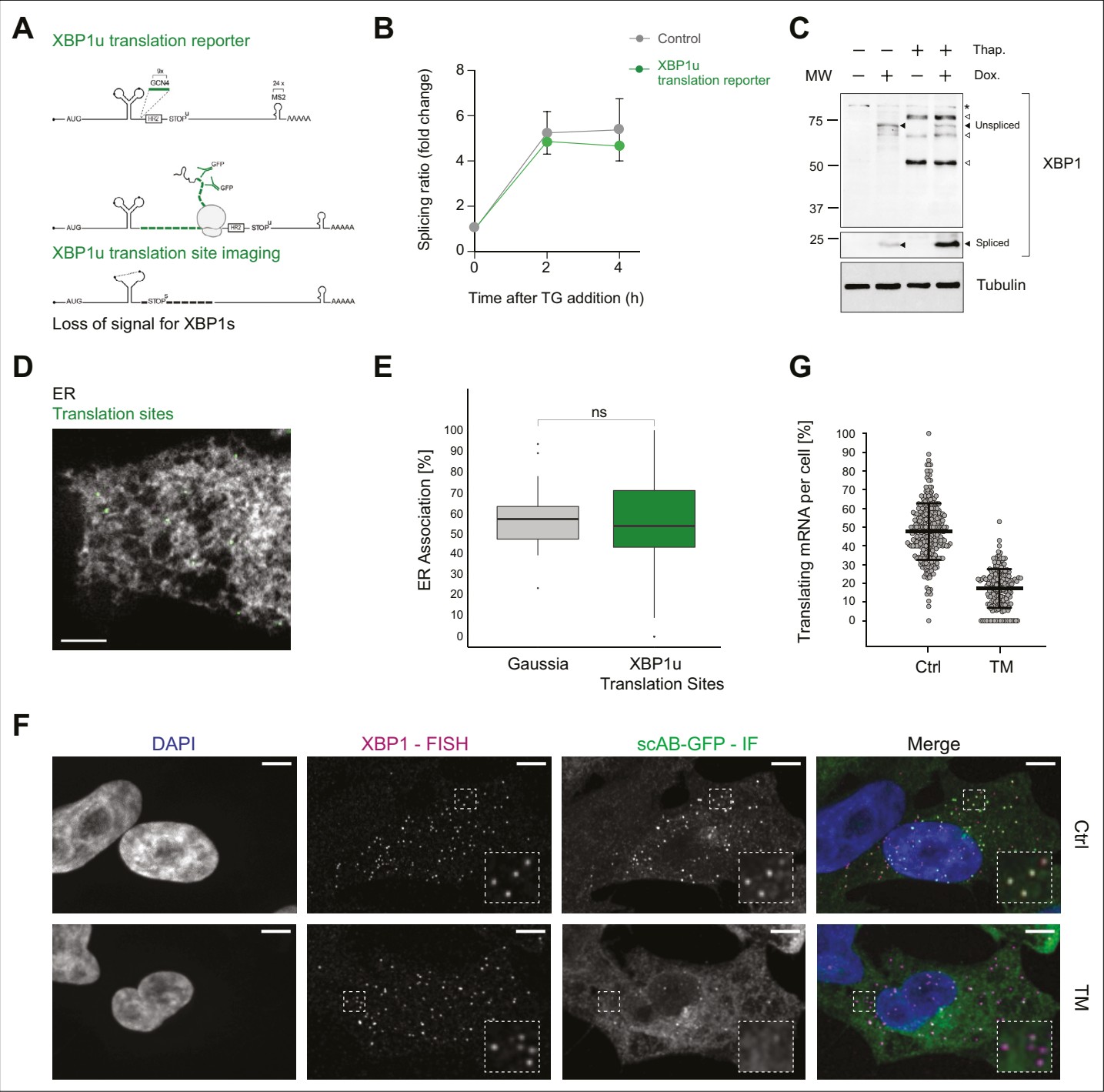

**Figure 2.** Association of XBP1u mRNA with the endoplasmic reticulum (ER) is translation dependent. (**A**) Reporter construct design and illustration of the method: XBP1u translation reporters feature a 9× GCN4 array (green) inserted into the opening reading frame downstream of the ER intron and in frame with the XBP1u protein. Upon translation of GCN4-XBP1u, emerging GCN4 peptide repeats are recognized by GFP-labeled single-chain antibodies (scAB-GFP), which allow detection of translating ribosomes together with mRNA transcripts. Upon splicing, the reading frame is changed and GCN4 expression is lost. (**B**) qPCR-based splicing assay to test functionality of XBP1u translation reporter (green) as compared to a non-GCN4-tagged control (gray). Shown is the splicing ratio (XBP1s/XBP1u) in response to induction of ER stress with 100 nM thapsigargin (TG). Graph represents the average ± SD (n=3). Statistical test Kruskal-Wallis and Dunn's multiple comparison test. No significant differences were observed.(**C**) Western blot against XBP1 proteins. Spliced XBP1 appearance is dependent on reporter expression (Dox) and induction of ER stress with 100 nM TG. Black arrows: XBP1 protein products expressed upon TG and Dox treatment. White arrows: unspecific bands present irrespective of reporter expression (Dox) in response to TG. Asterisk: unspecific bands present in all samples. (**D**) Representative live-cell image of XBP1u translation sites (green diffraction limited

*Figure 2 continued on next page*

*Figure 2 continued*

spots) in a HeLa cell expressing scAB-GFP and a fluorescent ER marker (gray). (**E**) Boxplot showing ER association of XBP1u translation sites (green) as compared to secreted protein encoding Gaussia mRNAs (gray) that serve as an ER-associated positive control. Statistical test: unpaired t-test, p-value = 0.49. (**F**) Combined single-molecule fluorescence in-situ hybridization (smFISH) and immunofluorescence (IF) analysis for colocalization of XBP1 mRNA (magenta) and translation site signal (green) in fixed HeLa cells (DAPI = blue). The majority of translation site spots disappear upon induction of ER stress with 5 μg/ml tunicamycin (TM) for 2 hours. (**G**) Quantification of data shown in (**F**). Individual dots represent per-cell averages. Black bars show mean ± SD. All scale bars = 5 μm. For raw data see *Figure 2—source data 1*.

The online version of this article includes the following source data and figure supplement(s) for figure 2:

**Source data 1.** Source data containing raw gel images for *Figure 2*.

**Figure supplement 1.** Live imaging of XBP1u translation sites.

**Figure supplement 2.** Colocalization control experiment shows no unspecific association of XBP1 mRNA and scAB-GFP spots.

Quantitative real-time (RT)-PCR as well as western blot analysis confirmed that this construct was able to undergo splicing upon induction of ER stress (*Figure 2B and C*). To characterize the translational status of XBP1 mRNA in live imaging experiments, we employed GFP-fused single-chain antibodies (scAB-GFP; *Voigt et al., 2017*; *Yan et al., 2016*) that specifically recognize GCN4 peptides and allow for detection of individual translation sites as diffraction-limited spots in the cytoplasm of HeLa cells co-expressing the Sec61b-SNAP ER marker (*Figure 2D*, *Figure 2—figure supplement 1A*, *Video 5*). As a further characterization of this experimental set-up, we performed a similar dual-color live imaging experiment but this time focused on the simultaneous detection of mRNA and translation site signals. To test whether the bright GFP signal corresponded to individual translation sites, we first acquired the NLS-stdMCP-stdHalo mRNA in parallel with the scAB-GFP translation site signal (*Figure 2—figure supplement 1B*) and then treated the cells with puromycin (PUR) to inhibit translation (*Figure 2—figure supplement 1C*). Upon PUR treatment, all scAB-GFP spots disappeared, which led us to conclude that they represent active translation sites.

We proceeded to quantify the degree of ER association observed for XBP1u translation sites in individual cells through the correlated diffusion and ER colocalization analysis introduced above (*Figure 2E*, *Figure 2—figure supplement 1A*). Interestingly, this analysis revealed that the majority of XBP1u translation sites colocalized with the ER (53.8 ± 22.1%, mean per cell ± SD) and exhibited a low mobility that is comparable to the behavior of predominantly ER-localized Gaussia transcripts (mean ER association = 57.3 ± 16.8%) but very different from the average degree of ER association assumed by XBP1 WT transcripts (mean ER association = 27.4 ± 19.4%). Thus, we conclude that XBP1u reporters are tethered to the ER surface in a translation-dependent manner.

As the translational frameshift induced by IRE1α-mediated splicing should abolish translation of the GCN4 repeats, we assessed the fraction of translating XBP1u transcripts in response to induction of ER stress. We treated the cells with TM and quantified the degree of colocalization for XBP1u mRNA and translation site spots. To maximize detection efficiency and more accurately estimate particle numbers per cell, we performed a combined single-molecule fluorescence in-situ hybridization (smFISH) and immunofluorescence (IF) experiment in fixed cells (*Figure 2F*, *Figure 2—figure supplement 1D*). Specifically, we used smFISH probes against the 5'-end of the *M. musculus* XBP1 ORF and an anti-GFP antibody that allowed for detection of the scAB-GFP labeled nascent polypeptides by IF. We confirmed that XBP1 mRNA and scAB-GFP translation site spots corresponded to single translation sites that only colocalized when expressed from the same mRNA transcripts (*Figure 2—figure supplement 2*).

Next, we investigated how ER stress affects the association of XBP1 mRNA with the ER and set out to determine how XBP1u molecules encounter IRE1α. In order to distinguish between the behavior of unspliced and spliced mRNA transcripts, we generated a XBP1 reporter variant with point mutations in the 5' and 3' splice sites

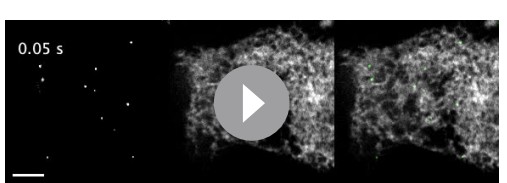

**Video 5.** Live imaging of XBP1u translation on the endoplasmic reticulum (ER). HeLa cell line stably expressing XBP1u translation reporter transcripts, scAB-GFP, and Sec61b-SNAP as ER marker. Simultaneous image acquisition for both channels (XBP1u translation sites, green, and ER, gray) using 50 ms exposure times (100 frames total). The movie is played at 20 fps. The scale bar is 5 μm.
https://elifesciences.org/articles/75580/figures#video5

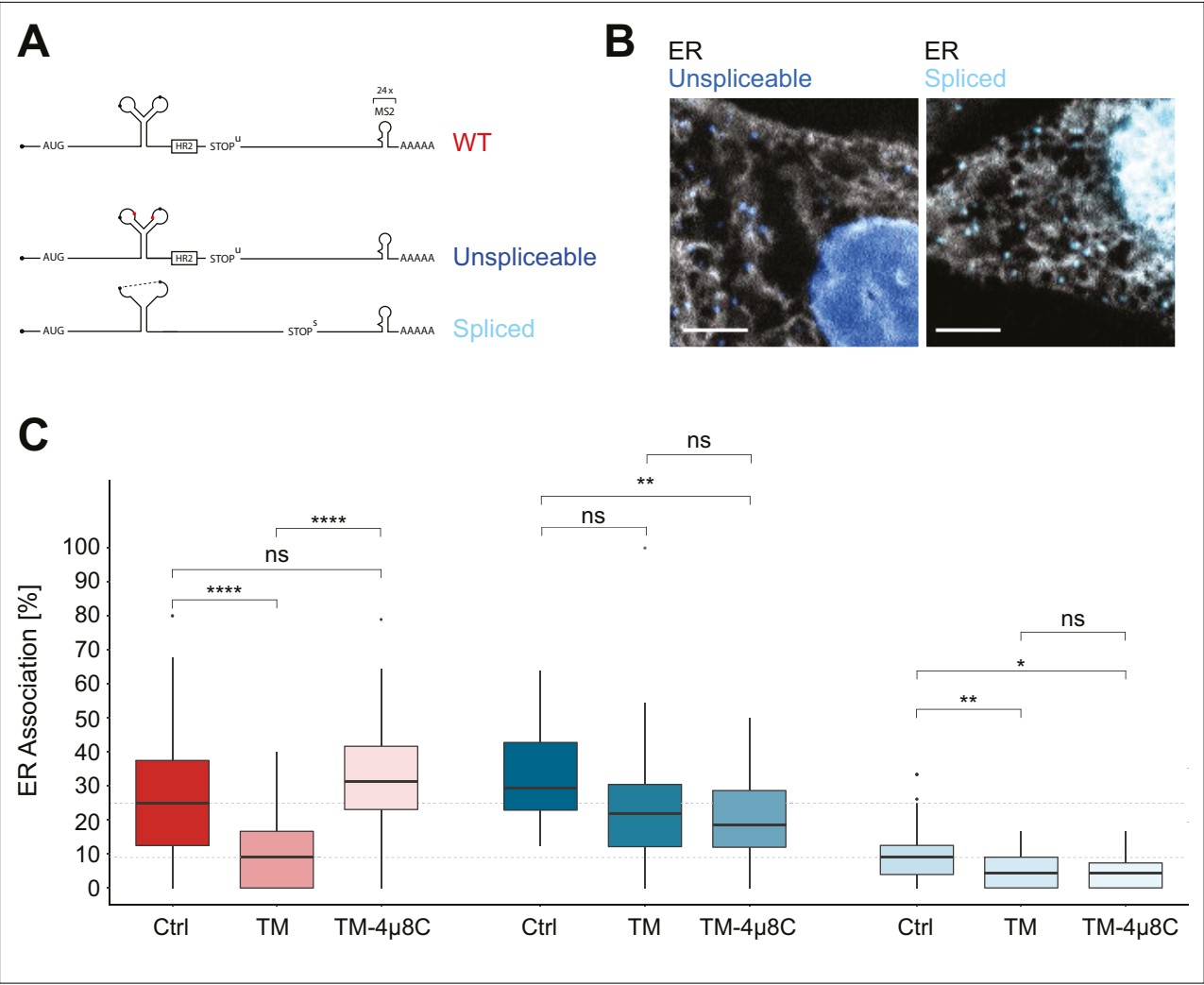

**Figure 3.** Inositol-requiring enzyme 1 alpha (IRE1α)-dependent processing and endoplasmic reticulum (ER) association of XBP1u transcripts during stress. (**A**) Reporter construct design: Unspliceable (dark blue) and spliced (light blue) reporter transcripts are identical to XBP1 wild-type (WT; red) except for point mutations in the intron (unsplicable) or complete lack of it (spliced). (**B**) Representative live-cell images of XBP1 splice site mutant reporters (blue) in HeLa cells expressing NLS-stdMCP-stdHalo and Sec61b-SNAP as ER marker (gray). (**C**) Boxplot showing quantification of ER association from correlated diffusion and ER colocalization analysis for XBP1 WT (red), unsplicable (dark blue), and spliced (light blue) reporter transcripts. Different opacities represent experimental conditions: no treatment (Ctrl), ER stress induced with 3–4 hours of 5 µg/ml tunicamycin (TM), ER stress induced with 3–4 hours of 5 µg/ml TM under IRE1α inhibition with 4µ8C (TM + 4µ8C). Statistical test: unpaired t-test, p-values: (p≥0.05)=ns; (p<0.0001)=****. (**D**) Representative live-cell images of XBP1 WT reporter constructs (red) in HeLa cells expressing NLS-stdMCP-stdHalo and Sec61b-SNAP as ER marker (gray) under ER stress (5 µg/ml TM) as well as ER stress with IRE1α inhibition (5 µg/ml TM and 50 µM 4µ8C). All scale bars = 5 µm.

The online version of this article includes the following source data and figure supplement(s) for figure 3:

**Figure supplement 1.** Validation of splice site mutants.

**Figure supplement 1—source data 1.** Raw gel images for *Figure 3—figure supplement 1*.

of the UPR intron that maintain its stem-loop structure but render the substrate cleavage incompetent (unsplicable, dark blue, *Figure 3A*, *Figure 3—figure supplement 1A, B*; *Calfon et al., 2002*; *Gonzalez et al., 1999*). In addition, we also generated a variant lacking the intron and constitutively expressing the XBP1s protein (spliced, light blue, *Figure 3A*, *Figure 3—figure supplement 1A, B*). We performed dual-color live imaging experiments (*Figure 3B*, *Videos 6 and 7*) and quantified reporter mobility and their degree of colocalization with the ER through correlated diffusion and ER colocalization analysis under non-stress conditions (*Figure 3—figure supplement 1C, D*). To further characterize particle behavior and control against potential higher-order oligomeric assemblies of reporter transcripts, we assessed the mean spot intensities detected for unsplicable and spliced

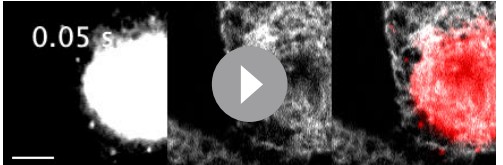

**Video 6.** Colocalization of XBP1 unspliceable mutant reporter transcripts with the endoplasmic reticulum (ER). HeLa cell line stably expressing XBP1 splice site mutant transcripts, NLS-stdMCP-stdHalo and an ER marker. Simultaneous image acquisition for both channels (XBP1 Unspliceable, blue and ER, gray) using 50 ms exposure times (100 frames total). The movie is played at 20 fps. The scale bar is 5 µm.
https://elifesciences.org/articles/75580/figures#video6

**Video 8.** Lack of colocalization with the endoplasmic reticulum (ER) exhibited by XBP1 WT transcripts in response to ER stress. HeLa cell line stably expressing XBP1 wild-type (WT) reporter transcripts, NLS-stdMCP-stdHalo, and an ER marker. Cells were treated with 5 µg/ml tunicamycin (TM) for 3–4 hours prior to image acquisition. Simultaneous image acquisition for both channels (XBP1 WT, red, and ER, gray) using 50 ms exposure times (100 frames total). The movie is played at 20 fps. The scale bar is 5 µm.
https://elifesciences.org/articles/75580/figures#video8

reporter transcripts. They were comparable to the mean intensities of XBP1 WT and HR2 mutant spots that were collected in the same imaging experiment (*Figure 3—figure supplement 1E*) and exhibited a single defined peak confirming that the spot signal originated from single mRNA transcripts that did not associate in larger oligomeric assemblies.

As expected, unspliceable reporter transcripts often exhibited a lower mobility (*Figure 3—figure supplement 1C*) and associated with the ER to a degree that is comparable to WT transcripts (*Figure 3C*) while spliced reporter mRNAs tended to diffuse at higher mobilities (*Figure 3—figure supplement 1D*) and displayed a lower degree of ER association (*Figure 3C*) comparable to cytoplasmic protein-encoding mRNAs (*Voigt et al., 2017*).

To determine the effect of ER stress on the ER association of WT, unspliceable and spliced reporter transcripts, we induced ER stress with TM (5 µg/ml) at least 3 hours before the start of the imaging session (*Video 8*). In addition, and to specifically assess the involvement of IRE1α in such association, we performed the same imaging experiments including 50 µM 4µ8C, a small molecule inhibitor that blocks substrate access to the active site of the IRE1α RNase domain and thereby selectively inactivates XBP1 cleavage (*Video 9*; *Cross et al., 2012*). As anticipated, ER stress-induced processing of WT reporters caused a strong decrease of their mean ER association from 27.4 ± 19.4% (mean ± SD) in the untreated condition to 10.1 ± 9.3% under TM treatment (*Figure 3C*, red). This result supports the idea that, upon completion of the splicing reaction, WT mRNAs are released from the ER membrane and behave like intron-free transcripts (*Figure 3C*, light blue) in the absence of ER stress (10.0 ± 9.1%). ER stress-induced mobilization of spliced WT reporter transcripts was a genuine consequence of IRE1α catalysis, since addition of 4µ8C to the TM condition restored ER association

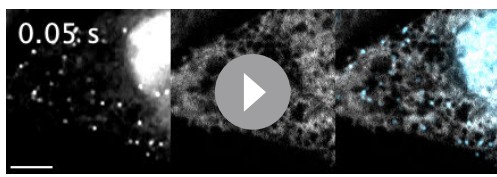

**Video 7.** Lack of colocalization with the endoplasmic reticulum (ER) exhibited by XBP1 spliced reporter transcripts. HeLa cell line stably expressing spliced XBP1 transcripts, NLS-stdMCP-stdHalo, and an ER marker. Simultaneous image acquisition for both channels (XBP1 spliced, light blue and ER, gray) using 50 ms exposure times (100 frames total). The movie is played at 20 fps. The scale bar is 5 µm.
https://elifesciences.org/articles/75580/figures#video7

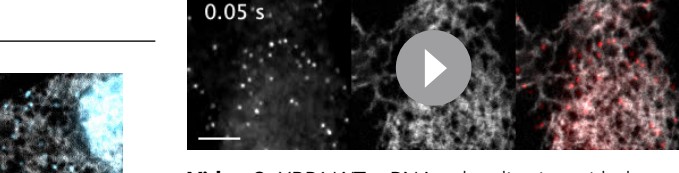

**Video 9.** XBP1 WT mRNA colocalization with the endoplasmic reticulum (ER) during ER stress and inhibition of inositol-requiring enzyme 1 alpha RNase activity. HeLa cell line stably expressing XBP1 wild-type (WT) reporter transcripts, NLS-stdMCP-stdHalo, and an ER-marker. Cells were treated with 5 µg/ml tunicamycin and 50 µM 4µ8C for 3–4 hours prior to image acquisition. Simultaneous image acquisition for both channels (XBP1 WT, red, and ER, gray) using 50 ms exposure times (100 frames total). The movie is played at 20 fps. The scale bar is 5 µm.
https://elifesciences.org/articles/75580/figures#video9

of WT reporters back to 33.2 ± 15.6% (*Figure 3C*, red). Taken together, these findings suggest that IRE1α-mediated splicing drives the release of translationally active, ER-tethered mRNAs.

In line with this notion, unspliceable reporter transcripts (*Figure 3C*, dark blue) not only associate with the ER to a high level (32.4 ± 13.5%) but also fail to show a similar reduction in ER association upon treatment with TM (23.7 ± 18.4%) and TM + 4μ8C (20.3 ± 12.6%) (*Figure 3C*). The same holds true for the spliced reporter construct. Since it does not encode the HR2 peptide and can therefore not be delivered to the translocon, it only associates with the ER to a limited extent (10.0 ± 9.1%). Upon induction of ER stress, its ER association rate is further reduced and similar to the unspliceable reporter, we do not observe significant changes in ER association between ER stress conditions in the absence (4.8 ± 5.0%) and presence (4.6 ± 4.4%) of 4μ8C.

We noticed that, for unspliceable and spliced reporters, ER stress caused a slight reduction of ER association when compared to untreated conditions. This effect may result from the general inhibition of cellular translation initiation triggered by the eIF2α kinase PERK, that promotes the UPR branch of the integrated stress response (*Pakos-Zebrucka et al., 2016*). It is plausible that the slightly reduced levels of ER association under ER stress conditions are due to the decreased recruitment of translating mRNPs to the ER surface, affecting all mRNAs to a limited extent (*Voigt et al., 2017*). This effect is not observed for the XBP1 WT reporter, where conversion of unspliced into spliced molecules is the major driver of mobilization from the ER. In summary, these experiments demonstrate that IRE1α activity is not required for ER association of XBP1 reporter mRNAs but suggest that IRE1α-mediated catalysis might contribute to the release of spliced mRNA molecules to the cytosol.

In combination, our findings support a model where IRE1α-mediated splicing is instrumental for the mobilization of XBP1 transcripts that are anchored to the ER in a translation-dependent manner. Based on this hypothesis, we sought to further investigate and visualize the sites of XBP1 processing on the ER membrane. To this aim, we developed an approach that allowed us to detect IRE1α in the reporter transcript-expressing HeLa cell lines introduced above and that was complementary to the single-molecule imaging approach, which detects endogenously tagged IRE1α protein molecules and is published in parallel to this study (*Belyy et al., 2021*). We knocked out the endogenous IRE1α using CRISPR/Cas9 and reconstituted its expression with a GFP-tagged IRE1α protein (*Figure 4A*). Identical to the previously published design of a splicing competent IRE1α-GFP construct, we introduced a GFP moiety in between the lumenal and kinase/RNase domains on the cytoplasmic site of the transmembrane protein (*Belyy et al., 2020*; *Li et al., 2010*).

IRE1α has been shown to form large oligomeric assemblies and microscopically visible clusters upon induction of ER stress in a number of studies (*Belyy et al., 2020*; *Kimata et al., 2007*; *Li et al., 2010*; *Tran et al., 2021*). Yet, the physiological relevance of these clusters remains unclear. To determine if the extent of IRE1α-GFP expression could artificially affect IRE1α clustering, we generated lentiviral constructs that induced different IRE1α-GFP expression levels. Specifically, we took advantage of the previously characterized Emi1 5'UTR that has been shown to downregulate translation approximately 40-fold (*Yan et al., 2016*). Since this 5'UTR was derived from the cell cycle protein Emi1, we termed the construct Emi1-IRE1α-GFP. For comparison, we also generated an IRE1α-GFP expression construct that was lacking the Emi1 5'UTR and expressed the IRE1α-GFP at higher levels.

As anticipated, western blot analysis confirmed that reconstituted IRE1α at low (Emi1-IRE1α-GFP) as well as at high levels (IRE1α-GFP) restored the functionality of IRE1α in KO cells, albeit to different extents (*Figure 4B*). While Emi1-IRE1α-GFP levels were similar to those of endogenous IRE1α, IRE1α-GFP expression was approximately 10-fold higher (*Figure 4C*). In both cell lines, ectopic IRE1α-GFP expression rescued XBP1 mRNA splicing under ER stress conditions, as determined by quantitative RT-PCR (*Figure 4D*) and by western blot detection of the resulting XBP1s protein (*Figure 4B*). At the same time, we noticed that GFP tagging slightly reduced XBP1 splicing levels both in the endogenous XBP1 mRNA and the WT reporter transcript (*Figure 4—figure supplement 1A-C* and *Figure 4D*). Other than that, IRE1α-GFP expressed from the Emi1 promoter supported transcription of XBP1-regulated genes, like ErdJ4, and did not significantly affect other signaling branches (*Figure 4—figure supplement 1D*). In line with previous reports (*Li et al., 2010*), strong overexpression of IRE1α-GFP triggered XBP1 mRNA splicing (and XBP1s synthesis) even in the absence of ER stress, underscoring the importance of adequate IRE1α expression levels for fine-tuning of the UPR.

In order to investigate if IRE1α clusters were the sites of XBP1 mRNA splicing on the ER, we imaged IRE1α-GFP in the HeLa cell lines stably expressing XBP1 reporter transcripts along with

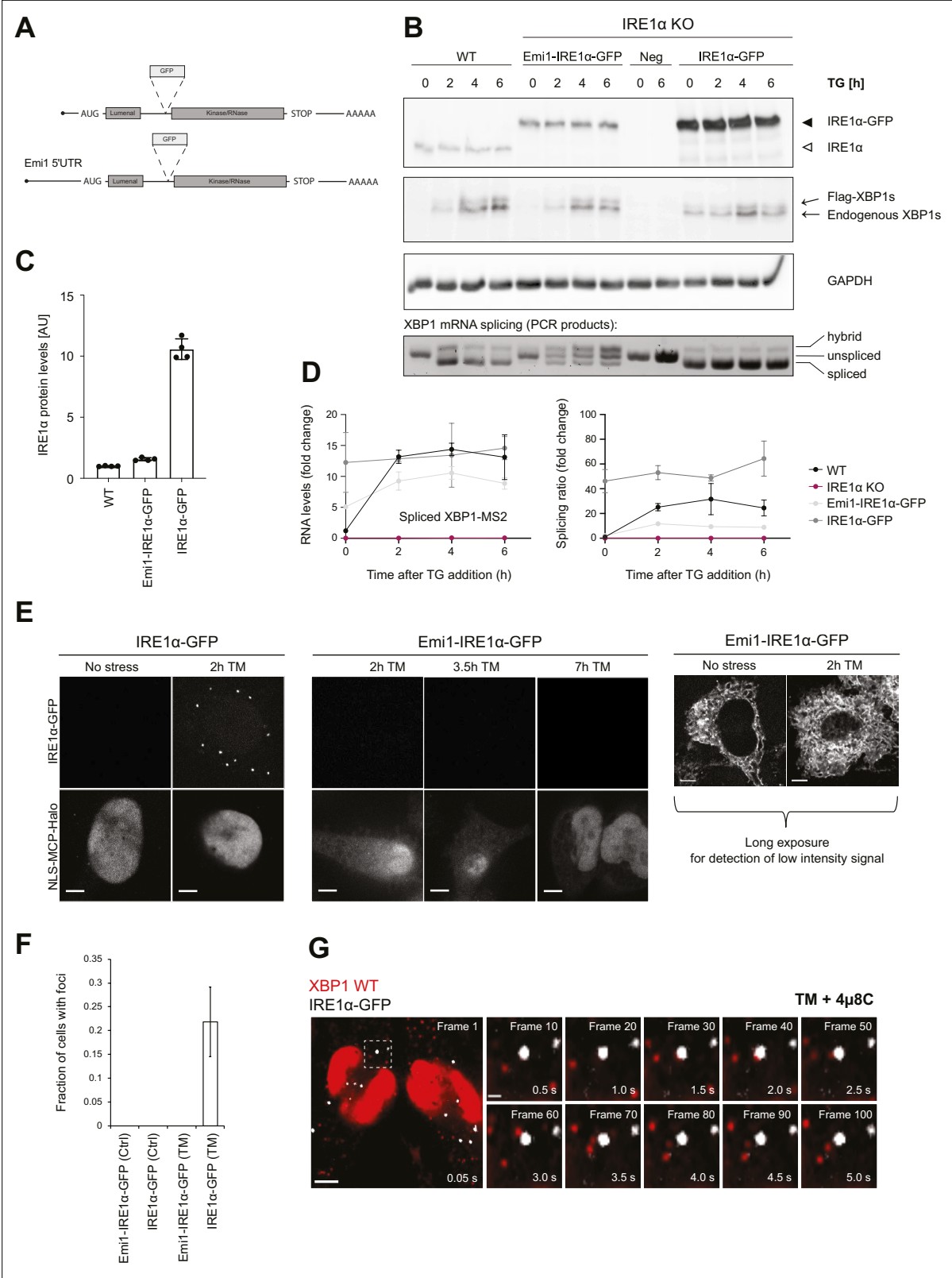

**Figure 4.** Inositol-requiring enzyme 1 alpha (IRE1α) is able to splice XBP1u mRNA in the absence of foci formation. (**A**) Schematic representation of IRE1α-GFP construct design analogous to *Belyy et al., 2020*. To reduce expression of IRE1α-GFP to match endogenous levels, part of the Emi1 5′ untranslated region (UTR) was inserted upstream of the IRE1α-GFP opening reading frame. (**B**) HeLa cells (wild-type [WT] or IRE1α knock-out) expressing either no IRE1α (Neg) or reconstituted IRE1α-GFP at low levels (Emi1-IRE1α-GFP) or at high levels (IRE1α-GFP) were kept untreated or treated with

*Figure 4 continued on next page*

*Figure 4 continued*

100 nM thapsigargin (TG) for indicated time points. Upper panels: western blot analysis of IRE1α and XBP1s levels in response to TG treatment. XBP1s immunodetection identifies two bands, a lower one corresponding to endogenous XBP1s and an upper one corresponding to the murine, FLAG-tagged XBP1s reporter protein. GAPDH (run in a different gel) was used as a loading control. Bottom panel: semiquantitative analysis of splicing of WT XBP1 mRNA. Total RNA was isolated from cells that were treated with TG as described above and subjected to RT-PCR with primers flanking the XBP1 intron. Lower band = spliced XBP1, middle band = unspliced XBP1, upper band = hybrid splicing intermediate (one strand spliced, one strand unspliced). (**C**) Quantification of the IRE1α expression levels in cell lines described in (**B**) under non-stress conditions. Graph depict the average ± SD (n=3). Revert staining of western blot membranes was used as a normalization value. (**D**) Quantitative RT-PCR to determine the levels of XBP1s mRNA and splicing ratios for the same RNA samples as shown in (**B**). Graph represents the average ± SD (n=3) (**E**) Representative live-cell images of the HeLa cell lines introduced in (**C**). In cells overexpressing IRE1α-GFP, foci can already be detected at 2 hour treatment with 5 µg/ml tunicamycin (TM). But there are no detectable IRE1α-GFP foci even after prolonged exposure to 5 µg/ml TM under standard imaging conditions in cells expressing Emi1-IRE1α-GFP. Only long exposure times allow for detection of low intensity GFP signal outlining the endoplasmic reticulum (ER) in the absence and presence of 5 µg/ml TM. (**F**) Quantification of the fraction of cells containing IRE1α-GFP foci in imaging cell lines under control (Ctrl) and ER stress (≥2 hour of 5 µg/ml TM) conditions. Cells are counted as foci-containing if ≥1% of the total cellular GFP signal is detected in IRE1α-GFP foci, which are defined as a ≥fivefold enrichment of GFP signal over cellular background. (**G**) Representative live-cell images of XBP1 WT reporters (red) in HeLa cells expressing NLS-stdMCP-stdHalo and IRE1α-GFP (gray) under ER stress (5 µg/ml TM) and IRE1α inhibition (50 µM 4µ8C). Dashed box indicates magnified inset and shows individual frames of the image series in the right part of the panel. The time series illustrates how individual mRNA particles (red) come close to IRE1α-GFP foci (gray) but do not associate stably nor accumulate in foci. All scale bars = 5 µm, except in single frame magnifications = 1 µm. For raw data see *Figure 1—source data 1*.

The online version of this article includes the following source data and figure supplement(s) for figure 4:

**Source data 1.** Raw gel images for *Figure 4*.

**Figure supplement 1.** Characterization of unfolded protein response (UPR) activation in Emi1-IRE1α-GFP expressing cells.

**Figure supplement 1—source data 1.** Raw gel images for *Figure 4—figure supplement 1*.

NLS-stdMCP-stdHalo and Sec61b-SNAP introduced above (*Figure 4E*). In agreement with earlier reports (*Li et al., 2010*) as well as a parallel study (*Belyy et al., 2021*), we detected IRE1α-GFP foci (defined as a ≥fivefold enrichment of GFP signal over background) in 21.89 ± 7.31% of cells expressing high levels of the fusion protein already after relatively short induction of ER stress with TM (5 µg/ml) for 2 hours (*Figure 4F*). Surprisingly, this was not the case for cells expressing Emi1-IRE1α-GFP at low levels, where we were unable to detect IRE1α-GFP clusters even after prolonged exposure to TM (5 µg/ml) for up to 7 hours. To make sure that we were not missing IRE1α clusters due to imaging conditions optimized for detection of fast-moving mRNA particles (e.g. short 50 ms exposure times), we acquired IRE1α-GFP signal from the same cells in the presence and absence of ER stress but this time using longer exposures (2000 ms) and maximum laser intensities. Under such conditions, we were able to detect IRE1α-GFP signal, which exhibited a characteristic ER-like distribution pattern but no IRE1α clusters (*Figure 4E*, right panel).

This observation suggested that IRE1α clusters are not necessary for the production of XBP1s, which we were able to detect in the absence of cluster formation (*Figure 4D*). To ensure that we were not missing a potential function of the previously observed IRE1α foci, we proceeded to image XBP1 WT mRNA recruitment to these oligomeric assemblies at high temporal and spatial resolution (*Figure 4F*, *Video 10*). Interestingly, we did not find XBP1 WT transcripts accumulating in IRE1α-GFP clusters even after prolonged TM treatment (5 µg/ml for up to 4 hours) and inhibition of IRE1α cleavage activity. XBP1 particles freely diffuse around IRE1α-GFP foci and only very rarely colocalize with the IRE1α-GFP signal (*Video 11*). This observation was true for both XBP1 WT (in the presence of 4µ8C) as well as unspliceable reporter transcripts (*Video 12*).

Taken together, our data indicate that Emi1-IRE1α-GFP supports splicing in the absence of foci formation. This suggests that XBP1 mRNA is spliced by lower oligomeric assemblies of IRE1α

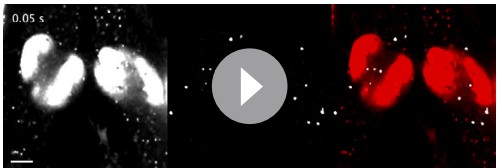

**Video 10.** No accumulation of XBP1 wild-type (WT) transcripts in inositol-requiring enzyme 1 alpha (IRE1α)-GFP foci during IRE1α inhibition. HeLa cell line stably expressing XBP1 WT reporter transcripts, NLS-stdMCP-stdHalo and IRE1α-GFP. Cells were treated with 5 µg/ml tunicamycin and 50 µM 4µ8C for 2–3 hours prior to image acquisition. Simultaneous image acquisition for both channels (XBP1 WT, red, and IRE1α-GFP, gray) using 50 ms exposure times (100 frames total). The movie is played at 20 fps. The scale bar is 5 µm.
https://elifesciences.org/articles/75580/figures#video10

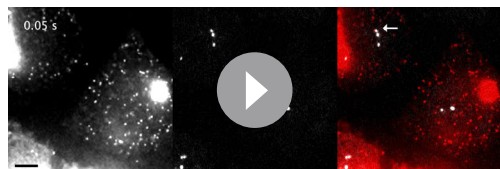

**Video 11.** Detection of single XBP1 wild-type (WT) transcripts in inositol-requiring enzyme 1 alpha (IRE1α)-GFP foci is possible but extremely rare. HeLa cell line stably expressing XBP1 WT reporter transcripts, NLS-stdMCP-stdHalo, and IRE1α-GFP. Cells were treated with 5 µg/ml tunicamycin and 50 µM 4µ8C for 2–3 hours prior to image acquisition. Simultaneous image acquisition for both channels (XBP1 WT, red, and IRE1α-GFP, gray) using 50 ms exposure times (100 frames total). The movie is played at 20 fps. The scale bar is 5 µm. White arrow indicates a single XBP1 mRNA particle that colocalizes with an IRE1α cluster. https://elifesciences.org/articles/75580/figures#video11

molecules, which can easily contact ER-associated ribosome-mRNPs, while IRE1α foci or large oligomeric clusters are not the sites of XBP1 mRNA processing during the UPR.

## Discussion

In this study, we present a single-molecule imaging approach that allows visualization of individual XBP1 transcripts and use it to investigate the recruitment of XBP1 mRNA to ER-localized IRE1α, which is a fundamental step of the XBP1 splicing mechanism. Based on previous yeast studies and the visualization of overexpressed IRE1α, splicing of XBP1 mRNA has been suggested to take place in large clusters of IRE1α oligomers that form during the UPR and could function as ER stress response centers (*Li et al., 2010*). Our findings challenge this view and suggest a different model for mammalian cells, where IRE1α might be recruited to the XBP1 mRNA and not vice versa.

Direct visualization of the recruitment of XBP1 mRNAs to the ER surface using single-molecule imaging revealed that XBP1 molecules become ER-associated in an HR2-dependent manner that is consistent with the targeting model proposed previously (*Kanda et al., 2016*; *Yanagitani et al., 2009*; *Yanagitani et al., 2011*). Furthermore, assessment of the translational status of single mRNA particles demonstrated that their ER association depends on interactions with the ribosome-nascent chain complex. Co-translational membrane tethering therefore immobilizes XBP1 transcripts on the ER surface and hints at a substrate recruitment mechanism where IRE1α diffuses through the ER membrane until it encounters XBP1u mRNAs at the Sec61 translocon. Direct interactions that have been reported for IRE1α and the translocon, SRP, as well as RNAs (*Acosta-Alvear et al., 2018*; *Plumb et al., 2015*) further increase the affinity of the interaction and underline the potential significance of such a recruitment mechanism.

Upon induction of ER stress, XBP1 transcripts are spliced and released from the ER surface. However, even though we can derive from our data that ER association correlates with IRE1α cleavage activity, we did not find XBP1 mRNAs colocalizing with IRE1α clusters. Moreover, large, microscopy-visible clusters were only detected when IRE1α-GFP was overexpressed at high levels. Thus, IRE1α foci are not the primary sites of XBP1 splicing. Instead, our findings support a model where ER-associated XBP1 transcripts are processed by small IRE1α oligomers that could dynamically assemble throughout the ER membrane.

These observations are in good agreement with an earlier study, in which the pharmacological activation of IRE1α with the flavonol luteolin promoted strong splicing of XBP1 in the absence of IRE1α clustering (*Ricci et al., 2019*). In addition, and directly related to our work, a parallel study shows that endogenously tagged IRE1α also fails to assemble into large clusters upon induction of ER stress (*Belyy et al., 2021*). In this work, the authors characterize IRE1α oligomerization during ER stress and find that the resting pool of IRE1α in the ER membrane is dimeric, while in response to stress transient IRE1α tetramers are assembled as the functional subunits that are required for trans-autophosphorylation and XBP1 splicing. Most likely, such a dynamic equilibrium

**Video 12.** No accumulation of XBP1 splice site mutant transcripts in inositol-requiring enzyme 1 alpha (IRE1α)-GFP foci. HeLa cell line stably expressing XBP1 splice site mutant reporter transcripts, NLS-stdMCP-stdHalo, and IRE1α-GFP. Cells were treated with 5 µg/ml tunicamycin for 3–4 hours prior to image acquisition. Simultaneous image acquisition for both channels (Unspliceable XBP1 reporter, blue, and IRE1α-GFP, gray) using 50 ms exposure times (100 frames total). The movie is played at 20 fps. The scale bar is 5 µm. https://elifesciences.org/articles/75580/figures#video12

between dimers and small oligomers allows cells to build timely, fine-tuned responses to local or transient perturbations in ER protein folding.

In combination, our findings suggest a novel mechanism for XBP1 recruitment to functional, dynamic IRE1α assemblies that continuously patrol the ER membrane to encounter substrates that are targeted there.

Following a different strategy, yeast IRE1p foci arrange the recruitment of unspliced *HAC1* mRNA to the ER membrane and efficiently localize the mRNA for splicing (*Aragón et al., 2009*; *van Anken et al., 2014*). Given the strong conservation of most UPR principles, upon visualization of large IRE1α clusters in human cells (*Li et al., 2010*; *Tran et al., 2021*) it was plausible to speculate that polarization of IRE1α might build splicing centers. However, these studies mostly relied on overexpression of ectopic IRE1α, which likely contributed to the perception that clusters were required for XBP1 splicing and explains the discrepancies between this and previous reports.

Our findings shed light on a fundamental step of the XBP1 splicing mechanism, which is the recruitment of IRE1α to ER-localized XBP1 transcripts. Yet, several open questions remain:

1. Are XBP1 mRNAs that are tethered to the ER surface as part of ribosome-nascent chain complexes continuously translated? Or is translation stably stalled while the mRNA remains poised for recruitment by IRE1α? And if so, how is translation resumed? And how relevant is the translational status of XBP1 mRNA for splicing?
2. Does IRE1α also bind XBP1 transcripts in the absence of the ribosome/translocon interaction? Since we observe a low degree of splicing for HR2-mutant reporters, we speculate that translation-independent recruitment of XBP1 transcripts and recognition through IRE1α might also be possible.
3. How does IRE1α discriminate between its distinct substrates? Beyond XBP1 splicing, IRE1α processes a broad range of substrates including RIDD mRNAs (*Hollien et al., 2009*; *Hollien and Weissman, 2006*) and a recently described, larger group of mRNAs that are processed through an unanticipated mode of cleavage with looser specificity (RIDDLE) (*Le Thomas et al., 2021*). Most of these mRNA substrates encode signal sequence-containing proteins and are delivered to the ER by SRP. While we know that activation of the IRE1α RNAse domain requires IRE1α dimerization/oligomerization as well as trans-autophosphorylation, the specific role and substrate specificity of the distinct assemblies remain unclear. It is tempting to speculate that the interplay of different oligomeric IRE1α assemblies with the translocon/ribosome/SRP environment may define the code for selective processing of distinct IRE1α substrates, avoiding the detrimental cleavage that might result from unrestrained RNA degradation.

In summary, our data have allowed us to uncover unanticipated features of one of the key steps of UPR initiation, the encounter of XBP1 mRNA with IRE1α to undergo splicing. Additional studies will be needed to further dissect the underlying mechanisms behind the regulation of IRE1α activity in homeostasis and disease.

## Materials and methods
### DNA constructs

The Gaussia luciferase reporter was the same as previously described (*Voigt et al., 2017*). Using the same plasmid backbone, we generated an XBP1 WT reporter, expressing an N-terminally FLAG-tagged *M. musculus* XBP1 coding sequence and 3' UTR (3'UTR covers nucleotides 1–948, considering +1 as the first nucleotide after the unspliced mRNA stop codon), followed by 24 MS2 stem loops. Nuclear introns were inserted into this construct to facilitate stability, nuclear export, and translation of the reporter mRNA.

HR2 mutant, spliced, and unspliceable constructs were generated by site-directed mutagenesis of WT RNA. In the HR2 mutant, one A nucleotide was inserted 45 nucleotides downstream the 3' splice site of murine XBP1. This insertion facilitates a translational frameshift that prevents HR2 synthesis, such that the amino acid sequence of the unspliced HR2 mutant protein is identical at the C-terminus to WT XBP1s. The spliced reporter plasmid is identical to WT but lacking the 26-nucleotide UPR intron. The unspliceable mutant bears point mutations at the 5' and 3' splice site loops. Almost invariant through evolution, positions 1, 3, and 6 of the splice site loops follow the consensus CNGNNGN (*Gonzalez et al., 1999*; *Hooks and Griffiths-Jones, 2011*). Mutation of either of these nucleotides

disrupts IRE1α cleavage in vitro and in vivo. Mutations in the 5' and 3' splice loops were tCGCAGC and CTaCAGC, respectively (mutation in lowercase).

For the translation reporter of XBP1u mRNA, a 9xGCN4 spaghetti monster (*Eichenberger et al., 2018*) was inserted 35 nucleotides downstream the 3' splice site, such that the spaghetti monster is in frame with the unspliced polypeptide. In this construct, we removed the last XBP1 nuclear intron, because the insertion of repeats in the close vicinity of its 5' splice site affected nuclear processing of the transcript.

We used KDEL-Turq2 (Addgene #36,204) and Sec61b-SNAP as fluorescent ER markers. Sec61b-SNAP was generated from Addgene construct #121,159 (GFP-Sec61b) through replacing the GFP with a SNAP moiety. Single-chain antibodies fused to GFP (scAB-GFP, Addgene #104,998) were used for imaging translation sites through nascent polypeptide labeling. NLS-stdMCP-stdHalo (Addgene #104,999) was employed for detection of single mRNA particles.

IRE1α-GFP was generated analogous to the construct design described by *Belyy et al., 2020*. The fusion protein includes a GFPuv tag (*Crameri et al., 1996*) and was integrated into a phage plasmid for lentiviral expression under the control of a constitutively active UbiC promotor. To reduce expression levels post-transcriptionally, the Emi1 5'UTR (*Yan et al., 2016*) was added upstream of the ORF.

## Cell line generation

HeLa cell lines stably expressing XBP1 and Gaussia luciferase reporter constructs were generated and maintained as previously described (*Voigt et al., 2017*). Briefly, reporter cassettes were stably integrated into parental HeLa 11ht cells that contain a single FLP site and express the reverse tetracycline-controlled transactivator (rtTA2-M2) for inducible expression (*Weidenfeld et al., 2009*). Cells were grown at 37°C and 5% $CO_2$ in DMEM + 10% FBS + 1% penicillin, streptomycin (Pen/Strep). Parental cells were authenticated via STR profiling (Eurofins). Mycoplasma contamination was regularly controlled for using mycoplasma detection PCR and smFISH.

IRE1 was knocked out by CRISPR/Cas9 editing of HeLa cells by transient transfection with the pX459v2-910 plasmid as in *Bakunts et al., 2017* kindly provided by Dr Eelco van Anken.

NLS-stdMCP-stdHalo, scAB-GFP, KDEL-Turq2, Sec61b-SNAP, IRE1α-GFP, and Emi1-IRE1α-GFP fusion proteins were stably integrated into the HeLa cell lines described above via lentiviral transduction. All cell lines were sorted using fluorescence-activated cell sorting to select for appropriate expression levels for single-molecule imaging.

## Western blots

For protein extraction in most experiments cells were treated with 100 nM TG (Sigma, stock 1 mM in dimethyl sulfoxide (DMSO)), HeLa cell monolayers were washed twice with ice-cold phosphate saline buffer (PBS), and then resuspended directly in Laemmli buffer, supplemented with protease and phosphatase inhibitors (Complete, Roche). Samples were heated to 95°C for 5 min, loaded on polyacrylamide gels (Thermo Fischer Scientific) and then transferred onto nitrocellulose (GE Healthcare). Successful protein transfer onto nitrocellulose was confirmed by reversible ponceau or revert staining. Immunoblot analysis was performed using standard techniques. All antibodies used in this study are listed in *Table 1*. Loading correction of immunoblot signals was performed by using GAPDH or tubulin signals as controls, or by quantifying revert fluorescence after transfer.

**Table 1.** List of antibodies used for western blotting.

| Protein | Provider | Cat. # | Notes |
|---|---|---|---|
| XBP1 | Santa Cruz Biotechnology | sc-7160 (M-186) | Detects murine XBP1 much better than endogenous, human XBP1 |
| XBP1 | Cell Signaling | #12,782 | Used to detect both human and murine XBP1 proteins |
| IRE1α | Cell Signaling | #3294 | |
| Calnexin | Novus Biologicals | NBP1-97485 | |
| Alpha-tubulin | Sigma | T6074 | |
| GAPDH | Cell Signaling | #2118 | |

**Table 2.** List of primers used for RT-PCR analysis.

| Oligonucleotides used in this study (1st Fwd; 2nd Rev.) |
| --- |
| H.s. Histone |
| AAAGCCGCTCGCAAGAGTGCG |
| ACTTGCCTCCTGCAAAGCAC |
| |
| H.s. GRP78 |
| GAGCTGTGCAGAAACTCCGGCG |
| ACCAACTGCTGAATCTTTGGAATTCGAGT |
| |
| H.s. XBP1u |
| CACTCAGACTACGTGCACCTC |
| CAGGGTGATCATTCTCTGAGGGGCTG |
| |
| H.s. XBP1s |
| CGGGTCTGCTGAGTCCGCAGCAG |
| CAGGGTGATCATTCTCTGAGGGGCTG |
| |
| M.m. XBP1u |
| CACTCAGACTACGTGCACCTC |
| CAGGGTGATCATTCTCTGAGGGGCTG |
| |
| M.m XBP1s |
| CGGGTCTGCTGAGTCCGCAGCAG |
| CAGGGTGATCATTCTCTGAGGGGCTG |
| |
| PCR to analyze M.m splicing by agarose electrophoresis |
| ACGCTGGATCCTGACGAGGTTCC |
| GAGAAAGGGAGGCTGGTAAGGAACTA |

Detection of immunolabeled proteins was performed using a commercial chemiluminescent assay (ECL prime; Amersham). Visualization and quantitative measurements were made with a CCD camera and software for western blot image analysis (Odissey Fc Imager System and Image Studio Lite v 4.0, respectively; Li-COR, Bad Homburg, Germany).

## RNA analysis

RNA extraction was performed using the guanidine isothyocyanate and phenol-chloroform method (TRIzol; Invitrogen). 1 µg of total RNA was treated with DNAse I and used for subsequent reverse transcription. 50–100 ng of total cDNA was used for RT-PCR using SybrGreen (BIORAD). RT-PCR primer sequences are listed in *Table 2*. For semi-quantitative assessment of splicing by PCR, we used primers flanking the XBP1 intron that specifically amplify murine but not endogenous human XBP1 mRNA. PCR products were resolved on 3% agarose gels.

## Membrane flotation assay

For flotation assays, we followed the method originally described by Mechler and Vassalli (*Mechler and Rabbitts, 1981*). 5 min before harvesting, subconfluent monolayers of cells were treated with 50 µg/ml cycloheximide (Sigma, Stock 50 mg/ml in DMSO) to prevent ribosomal runoff from mRNAs. Cultures were washed twice with chilled PBS and resuspended in hypotonic buffer medium (10 mM KCl, 1.5 mM MgCl$_2$, 10 mM Tris-HCl pH7.4, 50 µg/ml cycloheximide and protease and phospatase inhibitor) cocktail (Complete, Roche). Cells were allowed to swell for 5 min on ice, and then mechanically ruptured with a Dounce tissue grinder and spun for 2 min at 1000× g and 4°C. The supernatant was transferred to a new tube and supplemented with 2.5 M sucrose in TKM buffer (50 mM Tris-HCl pH7.4, 150 mM KCl, 5 mM MgCl$_2$, 50 µg/ml cycloheximide, protease and phosphatase inhibitors), to a final concentration of 2.25 M sucrose. This mixture was layered on top of 1.5 ml of 2.5 M-TKM in a SW40 polyallomer ultracentrifugation tube. On top of the extract-sucrose mix, we layered 6 ml of 2.05 M sucrose-TKM and 2.5 ml of 1.25 M sucrose-TMK. After centrifugation for 10 hours at 25,000 rpm in a SW40 Ti Beckman rotor, 1.5 ml fractions were collected from top to bottom and subjected to RNA and protein analysis.

## Single-molecule fluorescence in situ hybridization and immunofluorescence (smFISH-IF)

High precision glass coverslips (170 µm, 18 mm diameter, Paul Marienfeld GmbH) were placed into a 12-well tissue culture plate. 0.5 × 10$^5$ HeLa cells per well were seeded onto these cover slips and grown for 48 hours. Reporter expression was induced by addition of Dox (Sigma) to the medium for 2 hours. To ensure strong ER association phenotypes, Dox was removed from the medium after that

and cells were grown for another 2 hours until fixation. For ER stress conditions, 5 µg/ml TM (stock 5 mg/ml in DMSO, Sigma) was added at induction and maintained in the medium until fixation.

Combined smFISH-IF was performed as described previously (*Dave et al., 2021*). Briefly, single-molecule RNA detection was done using Stellaris FISH probes labeled with Quasar 570 (Biosearch Technologies) and designed against the 5' end of the *M. musculus* XBP1 ORF (*Table 3*). HeLa cells were washed with PBS twice and fixed with 4% paraformaldehyde (Electron Microscopy Sciences) for 10 min at room temperature (RT). This was followed by permeabilization in 0.5% Triton-X for another 10 min at RT. After two more washes with PBS, cells were preblocked in wash buffer (2 × SSC [Invitrogen], 10% v/v deionized formamide [Ambion], and 3% BSA [Sigma]) for 30 min at RT. Then hybridization buffer (150 nM smFISH probes, 2 × SSC, 10% v/v formamide, 10% w/v dextran sulphate [Sigma]) containing 1:1000 diluted anti-GFP antibody (Aves labs, GFP-1010) was added for 4 hours at 37°C. After hybridization, cells were washed with wash buffer twice for 30 min each, followed by incubation with antichicken IgY secondary antibody conjugated with Alexa-fluor 488 (1:1000 in PBS, Thermo Fisher, A-11,039) for 30 min. Coverslips were washed twice in PBS and then mounted on microscopy slides using ProLong Gold antifade reagent incl. DAPI (Molecular Probes).

smFISH-IF images were acquired on an inverted Zeiss AxioObserver7 microscope equipped with a Yokogawa CSU W1 scan head, a Plan-APOCHROMAT 100 × 1.4 NA oil objective, a sCMOS camera with chroma ZET405/488/561/647 nm emission filter and an X-Cite 120 EXFO metal halide light source. Z-stacks were acquired in 0.2 µm steps. Exposure times were 500 ms for Quasar 570 and 100 ms for the DAPI channel at maximum laser intensities while the IF signal was acquired at 20% 488 nm laser intensity for 200 ms.

## smFISH-IF data analysis

Detection of single mRNA and translation site spots from fixed cell imaging experiments was performed in KNIME (*Berthold et al., 2009*) as described previously (*Voigt et al., 2019a*; *Voigt et al., 2019b*).

Briefly, individual slices were projected as maximum intensity projections. mRNA and translation site spots were then separately detected using a custom-built KNIME node that runs the spot detection module of TrackMate (*Tinevez*

**Table 3.** List of smFISH probes to detect mouse XBP1 mRNA.

| 1 | taagagtagcactttggggg |
|---|---|
| 2 | gctactctgtttttcagttt |
| 3 | ctttctttctatctcgagca |
| 4 | ctgatttcctagctggagtt |
| 5 | cgtgagttttctcccgtaaa |
| 6 | tctggaacctcgtcaggatc |
| 7 | agaggtgcacatagtctgag |
| 8 | ttctggggaggtgacaactg |
| 9 | tgtcagagtccatgggaaga |
| 10 | actcagaatctgaagaggca |
| 11 | ccagaatgcccaaaaggata |
| 12 | aacatgacagggtccaactt |
| 13 | actctggggaaggacatttg |
| 14 | tggtaaggaactaggtcctt |
| 15 | gagttcattaatggcttcca |
| 16 | gcttggtgtatacatggtca |
| 17 | cagaggggatctctaaaact |
| 18 | acgttagtttgactctctgt |
| 19 | tgcttcctcaattttcacta |
| 20 | cctcttctgaagagcttaga |
| 21 | gagacaatgaattcagggtg |
| 22 | ttccaaaggctctttcttca |
| 23 | ccagctctgggatgaagtca |
| 24 | gctggatgaaagcaggtttg |
| 25 | caagaaggtggtctcagaca |
| 26 | atatccacagtcactgtgag |
| 27 | gtctgtaccaagtggagaag |
| 28 | cattggcaaaagtatcctcc |
| 29 | cactaatcagctgggggaaa |
| 30 | cagtgttatgtggctcttta |
| 31 | ctaggcaatgtgatggtcag |
| 32 | aagagacaggcctatgctat |
| 33 | cctctactttggcttttaac |
| 34 | ggaattcttctaaggccaga |
| 35 | cttggaagtcatctatgaga |
| 36 | ataccttagacagctgagtg |
| 37 | agctgtagtactggaatacc |
| 38 | tttagagtatactaccacct |
| 39 | aaactgtcaaatgaccctcc |

*Table 3 continued*

| | |
|---|---|
| **1** | **taagagtagcactttgggggg** |
| 40 | catgtccacctgacatgtcg |
| 41 | gaaatgctaagggccattca |
| 42 | cgaaacctgggaagcagaga |
| 43 | cataagggaaaacaagcccc |
| 44 | agatccatcaagcatttaca |

et al., 2017) in batch mode. This node is available in the KNIME Node Repository: KNIME Image Processing / ImageJ2 /FMI / Spot Detection (Subpixel localization). Detected mRNA and translation site spots in each channel were then colocalized using a nearest neighbor search to link mutual nearest neighbors between the two channels using a distance cut-off of three pixels. Nuclear segmentation was performed on the DAPI signal using the Otsu thresholding method while cytoplasmic segmentation was done using the smFISH background signal in the Q570 channel and a manual intensity threshold.

## Live-cell imaging

For live-cell imaging, cells were seeded in 35 mm glass-bottom μ-Dishes (ibidi GmbH) 48 hours prior to the experiment. Depending on the type of experiment, SNAP and Halo fusion proteins were labeled with JF549 or JF646 dyes (HHMI Janelia Research Campus) (*Grimm et al., 2015*) or SNAP-Cell Oregon Green (NEB, S9104S).

XBP1 mRNA expression was induced by addition of 1 μg/ml Dox to the medium. After 1–2 hours, Dox was removed to allow proper localization of XBP1 mRNAs to the ER membrane. To inhibit translation, cells were treated with 100 μg/mL PUR (stock 10 mg/ml in water, Invivogen) that was added to the cells directly prior to imaging. To induce ER stress, cells were treated with 5 μg/ml TM (stock 5 mg/ml in DMSO) that was added together with Dox at induction of mRNA expression and maintained in the imaging medium throughout the entire experiment. To inhibit IRE1α activity, the small molecule inhibitor 4μ8C (50 mM stock in DMSO, Sigma) was added at 50 μM together with Dox and maintained in the medium throughout the imaging experiment. Image acquisition was started not earlier than 1–2 hours after Dox removal to allow for localization of mRNA molecules and/or induction of the UPR.

Samples were imaged on an inverted Ti2-E Eclipse (Nikon) microscope equipped for live-cell imaging and featuring a CSU-W1 scan head (Yokogawa), two back-illuminated EMCCD cameras iXon-Ultra-888 (Andor) with chroma ET525/50 m and ET575lp emission filters, and an MS-2000 motorized stage (Applied Scientific Instrumentation). Illumination was achieved through 561 Cobolt Jive (Cobolt), 488 iBeam Smart, 639 iBeam Smart (Toptica Photonics) lasers, and a VS-Homogenizer (Visitron Systems GmbH). We used a CFI Plan Apochromat Lambda 100× Oil/1.45 objective (Nikon) that resulted in a pixel size of 0.134 μm. For all dual-color experiments, cells were imaged in both channels (single particles and ER) simultaneously and acquiring fast image series (20 Hz, 100 frames) in a single plane with two precisely aligned cameras. To correct for camera misalignment and chromatic aberrations, images of fluorescent TetraSpeck beads were acquired at each imaging session. Cells were maintained at 37°C and 5% $CO_2$ throughout the entire experiment.

## Correlated diffusion and ER colocalization analysis

Images of TetraSpeck beads were used to correct for the channel shift in affine transformation mode using the descriptor-based registration plugin (*Preibisch et al., 2010*) in Fiji (*Schindelin et al., 2012*). The transformation model obtained after aligning the bead images was then reapplied to translate the single mRNA/translation site channel onto the ER channel using as custom-made Fiji macro (*Mateju et al., 2020*).

Single-particle diffusion and ER colocalization analysis were performed as described previously (*Voigt et al., 2017*). Briefly, we used the KNIME analytics platform (*Berthold et al., 2009*) and a data processing workflow that allows for segmentation of the ER signal through trainable pixel classification using ilastik (*Berg et al., 2019*). The resulting probability maps are transformed to binary images, which are in turn used to generate distance maps that attribute intensity values to each pixel position with respect to its distance to the closest ER boundary. Positions on the ER are given positive values, while positions away from the ER are defined as negative values. The workflow further correlates mRNA positions (X and Y coordinates) obtained from SPT to the ER boundaries at any time point throughout the experiment and computes a cumulative ER localization index through addition of all

**Table 4.** Imaging data statistics.

Data statistics for live imaging experiments

| Reporter | Experiment | Independent replicates | ID experiments | Cells | Mean tracks (≥3 frames) per cell | Tracks ≥3 frames | Tracks ≥10 frames | Tracks ≥30 frames |
|---|---|---|---|---|---|---|---|---|
| Gaussia | Ctrl | 2 | 20200323, 20200420 | 19 | 149 | 2822 | 1480 | 864 |
| XBP1 wt | Ctrl | 4 | 20191023, 20191108, 20200313,20210702 | 37 | 141 | 5200 | 2588 | 997 |
| XBP1u translation reporter | Ctrl | 3 | 20211018, 20211022, 20211023 | 50 | 36 | 1815 | 923 | 519 |
| XBP1 HR2 mutant | Ctrl | 3 | 20191023, 20191108, 20200313 | 25 | 206 | 5144 | 2454 | 653 |
| Spliced | Ctrl | 3 | 20200313, 20210226, 20211022 | 36 | 193 | 6943 | 3261 | 955 |
| Unspliceable | Ctrl | 3 | 20200313, 20210226, 20211025 | 34 | 194 | 6612 | 2997 | 1120 |
| XBP1 wt | TM | 3 | 20210218, 20210322, 20210702 | 37 | 117 | 4326 | 2183 | 741 |
| Spliced | TM | 3 | 20210218, 20210322, 20211022 | 35 | 152 | 5309 | 2571 | 817 |
| Unspliceable | TM | 3 | 20210218, 20210322, 20211025 | 43 | 120 | 5143 | 2491 | 1001 |
| XBP1 wt | TM +4μ8C | 3 | 20210219, 20210702, 20211017 | 41 | 119 | 4896 | 2509 | 1046 |
| Spliced | TM +4μ8C | 3 | 20210219, 20211017, 20211022 | 35 | 173 | 6,044 | 2,792 | 793 |
| Unspliceable | TM +4μ8C | 3 | 20210219, 20211017, 20211025 | 48 | 120 | 5750 | 2766 | 1031 |

**Data statistics for smFISH experiments**

| Condition | Experiment | | Colocalization control | |
|---|---|---|---|---|
| Condition | Ctrl | TM | XBP1 wt +Renilla Luciferase | XBP1u only |

*Table 4 continued on next page*

## Table 4 continued

**Data statistics for live imaging experiments**

| | | | | |
|---|---|---|---|---|
| Cells | 278 | 170 | 208 | 114 |
| Replicates | 2 | 2 | 1 | 1 |
| scAB-GFP spots | 4172 | 1928 | 4086 | 1928 |
| XBP1 mRNA spots | 6704 | 4186 | 5230 | 2313 |
| Mean fraction of transl. mRNAs | 0.47 | 0.18 | 0.011 | 0.498 |

**Data statistics for IRE1a-GFP foci quantification**

| Experiment | Construct name | Replicate 1 | Replicate 2 | Replicate 3 | Replicate 4 |
|---|---|---|---|---|---|
| TM | Emi1-IRE1α-GFP | 20220510_577 | 20220510_630 | 20220513_630 | |
| | Cells total | 233 | 115 | 167 | |
| | Cells w/o foci | 233 | 115 | 167 | |
| | Cells with foci | 0 | 0 | 0 | |
| | Mean(Fraction in foci) | 0 | 0 | 0 | |
| | SD (Fraction in foci) | 0 | 0 | 0 | |
| | Fraction(Cells with foci) | 0 | 0 | 0 | |
| | Image series | 10 | 10 | 10 | |
| | IRE1α-GFP | 20200826_631 | 20220510_630 | 20220513_630 | |
| | Cells total | 55 | 178 | 301 | |
| | Cells w/o foci | 43 | 152 | 213 | |
| | Cells with foci | 12 | 26 | 88 | |
| | Mean(Fraction in foci) | 0.01920605 | 0.02054973 | 0.02550403 | |
| | SD (Fraction in foci) | 0.00850352 | 0.01393375 | 0.01711448 | |
| | Fraction(Cells with foci) | 0.21818182 | 0.14606742 | 0.2923588 | |

*Table 4 continued on next page*

*Table 4 continued*

Data statistics for live imaging experiments

**Ctrl**

**Emi1-IRE1α-GFP**

| Image series | 8 | 10 | 15 | |
|---|---|---|---|---|
| | 20220427_630 | 20220428_577 | 20220428_630 | 20220513_630 |
| Cells total | 76 | 107 | 53 | 136 |
| Cells w/o foci | 76 | 107 | 53 | 136 |
| Cells with foci | 0 | 0 | 0 | 0 |
| Mean(Fraction in foci) | 0 | 0 | 0 | 0 |
| SD (Fraction in foci) | 0 | 0 | 0 | 0 |
| Fraction(Cells with foci) | 0 | 0 | 0 | 0 |
| Image series | 5 | 5 | 5 | 10 |

**IRE1α-GFP**

| | 20220427_577 | 20220428_577 | 20220428_630 | 20220513_630 |
|---|---|---|---|---|
| Cells total | 52 | 46 | 123 | 105 |
| Cells w/o foci | 52 | 46 | 123 | 105 |
| Cells with foci | 0 | 0 | 0 | 0 |
| Mean(Fraction in foci) | 0 | 0 | 0 | 0 |
| SD (Fraction in foci) | 0 | 0 | 0 | 0 |
| Fraction(Cells with foci) | 0 | 0 | 0 | 0 |
| Image series | 5 | 4 | 5 | 5 |

intensity values that correspond to the positions assumed by a transcript over the experimental time course. To obtain a measure for particle mobility, the workflow further determines IDCs for each track. These are calculated as the mean of all displacements measured by SPT over 100 frames (*Berg, 1993*) and can be computed by a custom-made component node that is also available from the KNIME hub (KNIME Hub >Users > imagejan >Public > fmi-basel >components > Instantaneous diffusion coefficient).

ER association was quantified for all particles that could be tracked for at least 30 frames and was performed based on IDCs and cumulative ER colocalization indices as described before (*Voigt et al., 2017*). Values were plotted as scatter plots using the ggplot2 package in R. For the quantification of the degree of ER association per cell, only cells including at least three tracks were included. The analysis was also performed in KNIME and box plots were generated using the ggplot2 and ggpubr packages in R. Data overview and statistics for all live imaging experiments are summarized in *Table 4*.

## Data and software availability

All data were analyzed using custom-made KNIME image analysis workflows that have been published before (*Voigt et al., 2017*). Specialized KNIME component nodes are available from the KNIME Hub (Users > imagejan > Public > fmi-basel). To prepare exemplary microscopy data for publication, image series were processed in Fiji (*Schindelin et al., 2012*). Manuscript figures were prepared using Adobe InDesign and Illustrator 2021. All processed and raw microscopy data generated in this study are available from Zenodo (*Gómez-Puerta et al., 2022a*; *Gómez-Puerta et al., 2022b*; *Gómez-Puerta et al., 2022c*; *Gómez-Puerta et al., 2022d*; *Gómez-Puerta et al., 2022e*). All other raw data, including full gel images have been uploaded as source data to this manuscript.

## Acknowledgements

This work was funded by the Ministerio de Ciencia e Innovación (PID2020-120497RB-I00/financiado por MCIN/ AEI /10.13039/501100011033) (TA), the Novartis Research Foundation (JAC), and a Boehringer Ingelheim Fonds PhD fellowship (TH). The authors thank L Gelman, L Plantard, and J Eglinger (FMI) for microscopy and image analysis support and H Kohler (FMI) for cell sorting. We thank Urs Greber and Maite Huarte for critical reading of the manuscript and all members of the Chao and Aragón labs for their input and support.

## Additional information

### Funding

| Funder | Grant reference number | Author |
| --- | --- | --- |
| Ministerio de Ciencia, Innovación y Universidades | PID2020-120497RB-I00 | Tomás Aragón |
| Boehringer Ingelheim Fonds | PhD fellowship | Tobias Hochstoeger |

The funders had no role in study design, data collection and interpretation, or the decision to submit the work for publication.

### Author contributions

Silvia Gómez-Puerta, Roberto Ferrero, Tobias Hochstoeger, Ivan Zubiri, Investigation; Jeffrey Chao, Funding acquisition; Tomás Aragón, Conceptualization, Data curation, Formal analysis, Funding acquisition, Investigation, Methodology, Project administration, Supervision, Validation, Visualization, Writing – original draft, Writing – review and editing; Franka Voigt, Conceptualization, Data curation, Formal analysis, Investigation, Methodology, Project administration, Resources, Software, Supervision, Validation, Visualization, Writing – original draft, Writing – review and editing

### Author ORCIDs

Tobias Hochstoeger ![ORCID] http://orcid.org/0000-0002-8061-7857
Tomás Aragón ![ORCID] http://orcid.org/0000-0002-1700-2729

Franka Voigt [ORCID] http://orcid.org/0000-0001-9515-0367

Decision letter and Author response
Decision letter https://doi.org/10.7554/eLife.75580.sa1
Author response https://doi.org/10.7554/eLife.75580.sa2

## Additional files

### Supplementary files
• Transparent reporting form

### Data availability
All imaging data generated in this study have been uploaded to Zenodo.

The following datasets were generated:

| Author(s) | Year | Dataset title | Dataset URL | Database and Identifier |
|---|---|---|---|---|
| Gómez-Puerta S, Ferrero R, Hochstoeger T, Zubiri I, Chao J A, Aragón T, Voigt F | 2022 | Live imaging data of XBP1 mRNA under control conditions | https://doi.org/10.5281/zenodo.6559201 | Zenodo, 10.5281/zenodo.6559201 |
| Gómez-Puerta S, Ferrero R, Hochstoeger T, Zubiri I, Chao J A, Aragón T, Voigt F | 2022 | Live imaging data of XBP1 mRNA under ER stress conditions | https://doi.org/10.5281/zenodo.6557623 | Zenodo, 10.5281/zenodo.6557623 |
| Gómez-Puerta S, Ferrero R, Hochstoeger T, Zubiri I, Chao J A, Aragón T, Voigt F | 2022 | Live imaging data of XBP1 mRNA under ER stress and IRE1a inhibition | https://doi.org/10.5281/zenodo.6559058 | Zenodo, 10.5281/zenodo.6559058 |
| Gómez-Puerta S, Ferrero R, Hochstoeger T, Zubiri I, Chao J A, Aragón T, Voigt F | 2022 | Live cell imaging data of IRE1a-GFP foci formation | https://doi.org/10.5281/zenodo.6559125 | Zenodo, 10.5281/zenodo.6559125 |
| Gómez-Puerta S, Ferrero R, Hochstoeger T, Zubiri I, Chao J A, Aragón T, Voigt F | 2022 | Fixed cell imaging data of XBP1 mRNA | https://doi.org/10.5281/zenodo.6559109 | Zenodo, 10.5281/zenodo.6559109 |

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
