## [Editor Report]

We agree that this study, especially when considered in parallel with the work from Belyy et al., significantly furthers our understanding of how early events in the unfolded protein response pathway trigger downstream signals. This pathway is essential to respond and protect against potentially toxic insults to ER homeostasis. On a more general note, the advances in single-molecule optical imaging, which were developed for your work, will benefit others who wish to probe dynamic signaling events at the ER membrane and beyond.

---

## [Decision Letter]

**Decision letter after peer review:**

[Editors’ note: the authors submitted for reconsideration following the decision after peer review. What follows is the decision letter after the first round of review.]

Thank you for submitting the paper "Live imaging of the co-translational recruitment of XBP1 mRNA to the ER and its processing by diffuse, non-polarized IRE1α" for consideration by *eLife*. Your article has been reviewed by 3 peer reviewers, and the evaluation has been overseen by a Reviewing Editor and a Senior Editor. The reviewers have opted to remain anonymous.

Comments to the Authors:

While there was significant interest in the topic and preliminarily findings, we are sorry to say that, after consultation with the reviewers, we have decided that this work will not be considered further for publication by *eLife*.

As you will read below, whilst all three reviewers concur with the importance of the paper's subject matter they independently concluded that the study would need significant additional work to become suitable for publication. The reviewers have laid out a non-trivial number of additional controls and experiments that are considered essential. In brief, the absence of these experiments indicates that many of the claims currently lack experimental support to the level required by *ELife*. More specifically, the clustering analysis is of limited scope and significance, temporal aspects of the observed events are absent, the tools used for this study need to be better validated, and single particle analysis was--in contrast to what was stated--not performed rigorously. There is also a consensus amongst the reviewers that a substantial portion of the paper is simply confirmatory based on prior work performed by the Kohno and Argon labs.

*Reviewer #1 (Recommendations for the authors):*

In this study, the authors explore key steps in the mammalian unfolded protein response (UPR). They focus on the IRE1a branch, specifically, the recruitment of target mRNA XBP1 to the endoplasmic reticulum (ER) surface, the fate of the mRNA, and organization of IRE1a, itself, in response to unfolded protein stress. The novelty of this study is the use of live cell imaging techniques to follow XBP1 mRNA localization and dynamics. The translational recruitment model of XBP1 to the ER membrane surface has been previously described by others and is confirmed in live cells, here. Different reporters were used to visualize localization and translation of spliced and unspliced forms of XBP1. The authors make additional claims regarding the stress-induced clustering behavior of IRE1a. Overall, this study adds visual data, with some dynamics information to potentially help refine models of steps of XBP1 mRNA trafficking and splicing in response to ER stressors. The novelty of the study approach is undercut by the need for more thorough characterization of the reporters, better descriptions of the reagents, and more accurate descriptions of experiments and results. With the additional requested details and modifications to the text, this manuscript could be a useful addition to the ER stress literature and of broader interest to groups interested in trafficking and translation on the ER membrane.

The authors make two claims that are well supported by imaging data, mutants, and biochemical fractionation.

Splicing activity by IRE1a or the presence or expression of XBP1 message in an already spliced form decreases the fraction of XBP1 associated with the ER. These observations are supported by membrane fractionation experiments.

The authors have confirmed that stable ER association of unspliced XBP1 mRNA is established through HR2-dependent targeting and relies on active translation. The experimental data support this claim.

Other claims are less well supported or less accurate.

First, the authors claim to "directly monitor recruitment of individual XBP1 transcripts to the ER surface." I was unable to find a single movie or data series demonstrating the movement of a single particle from the cytoplasm to the ER membrane. Experiments show mRNAs on the ER or report amount of cytoplasmic vs. membrane bound. The results, presumably, of recruitment are observed and reported, not the process.

The authors state "Next, we quantified the mobility of individual particles with respect to their ER localization and therefore assessed when an mRNA particle associates with the ER." What appears to be assessed is whether a particle is associated with the ER, not "when." I assume the authors did not mean "when" as part of temporal order, but it would be less confusing to claim to assess whether or if.

The authors claim to show that IRE1a-catalyzed splicing mobilizes XBP1 mRNA from the ER membrane in response to ER stress. Here the data are mostly supportive of the claim. However, what the authors actually show is that splicing is necessary for an increase in XBP1 mRNA localization in the cytoplasm. No data are presented showing interactions with IRE1a and/or movement of XBP1 mRNA from the ER membrane to the cytoplasm. Based on multiple studies, the interpretation is likely correct, but the microscopy data do not show the process.

The matter of IRE1a clustering is more problematic.

The authors state, "Surprisingly, we find that XBP1 transcripts are not recruited into large IRE1a clusters, which only assemble upon overexpression of fluorescently-tagged IRE1a during ER stress. Our findings support a model where ribosome-engaged, ER-poised XBP1 mRNA is processed by functional IRE1a assemblies that are homogenously distributed throughout the ER membrane."

The authors state, "ER-poised XBP1 transcripts are processed by functional IRE1a assemblies that are homogenously distributed throughout the ER membrane." No data are provided on what fraction of the IRE1a is in an active state. There is no method used to detect the active state or information on the spatial distribution of stress-activated IRE1a. This claim is not supported.

"Our data have allowed us to visualize and uncover unanticipated features of one of the key steps of UPR initiation, the encounter of XBP1 mRNA with IRE1a to undergo splicing." This is another claim that is not quite accurate. The authors observed tagged XBP1 mRNA dynamics in co-expressing cells treated with an inhibitor of IRE1a activity. An encounter in which splicing could actually occur is never presented. Whether XBP1 mRNA is stimulated to dissociate from the ER membrane following an encounter with IRE1a-GFP is not presented. The ability of drug inhibited IRE1a-GFP to bind XBP1 mRNA is not tested or reported.

The authors have made a significant point of weighing in on the matter of IRE1a clustering in both the abstract and text. While their findings are potentially interesting, the investigation of this matter has not been sufficiently rigorous. First, the authors simply describe the attached fluorescent protein as a GFP. Is this monomeric GFP? Emerald? mNeonGreen? or? With the wide variety of fluorescent proteins available, the minimal expectation is to state what was inserted and imaged. Given the potential of many fluorescent proteins to oligomerize, the choice of fluorescent protein matters.

Second, the systems used may be inducing some constitutive stress. For example, in 2C, there appears to be splicing of the reporter, even in the absence of thapsigargin or expression of IRE1-fluorescent protein. Induction with Dox is for 15 hours. That's long enough for stimulation of a stress response and perhaps even some degree of adaptation. I'd like to see some other assays of ER stress before and during induction up to the time normally used to apply stress. Similarly, in the Ire1a-KO cells in 4B, there appears to be spliced XBP1, with and without stress. Importantly, the ratio of spliced XBP1 for the emi1 variant is much lower than for wt, even though immunoblotting suggests there is significantly more EMI-1-IRE1a-GFP protein present than for wt IRE1a. Please comment. Is the GFP tagged variant less active? Is it possible that the GFP tagged variant is less activatable?

More fundamentally, the authors claim that large IRE1a clusters… only assemble upon overexpression of fluorescently-tagged IRE1a during ER stress. At the very least, the more accurate thing would be to say fluorescently-tagged IRE1a only forms inducible observable clusters under overexpression conditions. In this study, there has been no testing of oligomeric/cluster status of native IRE1a. There has been no use of other assays to determine if low expressed IRE1a-fluorescent protein forms oligomers of any size during stress. Nor has there been investigation of whether a version with a small epitope tag might form clusters. What about the ability of other stress conditions, such as acute thapsigargin or DTT, for the ability to induce observable clustering for the lower expressed IRE1a-GFP? Ultimately, the clustering analysis has been too limited to draw any major conclusions. The ability to cluster and its relevance are important to popular models of IRE1a activity. A nuanced study of the clustering regulation of IRE1a would be immensely useful to the field. The observation that clustering observed in mammalian cells is stress inducible, suggests that observable clustering is probably reporting on some aspect of IRE1 regulation. Clusters may represent an extreme manifestation of normal physiology. That is, even an artifact of expression does not rule out the potential importance of forming small clusters in the normal stress response. Note that the overexpressed IRE1a-GFP is constitutively maximally active. There is no stress induced increase in splicing activity, yet there is a stress-inducible change in distribution. This redistribution could reflect some sort of stress inducible regulatory step, for example sequestering of excess unengaged IRE1a, inactivated IRE1a or something else. Whatever causes visible clustering seems to be more than a simple artifact.

It would be helpful for the authors to compare the functionality of their constructs/system relative to untagged versions. Specifically, rates of processing. While I think the authors have demonstrated reasonably well that their XBP1 MS2 reporter does get targeted to the ER and spliced in response to ER stress, it would be useful to know if the 24 MS2 sites and attached MCS-GFP reporters (5nm+ diameter for each) affect rates of XBP1 mRNA processing. Is it possible that the construct is so large that it is relatively immobilized on the ER membrane by sterics and/or sheer size? Could the size of the reporter slow the approach of IRE1a to the cleavage site? Does high expression of the XBP1 substrate impact the efficiency of processing? That is, if client is saturating, then encountering and splicing by activated IRE1a on the ER membrane could compensate for potential issues due to size or mobility. In addition, it would be useful to determine how much XBP1 reporter mRNA is expressed and how this compares to endogenous levels of XBP1 mRNA in unstressed and stressed cells.

*Reviewer #2 (Recommendations for the authors):*

This manuscript develops a different reporters to monitor XBP1 targeting to the ER, which are used to confirm previous results showing that XBP1 is directed to the ER through a mechanism involving translation of the HR2 mRNA sequence. As indicated in the manuscript, this mechanism had been previously reported by Kohno, and, while the work presented here confirms this model, it does not extend it. The major advance from this manuscript, apart from the reporter development, relates to the fact that IRE1 clusters are not observed in cells expressing endogenous levels of IRE1-GFP and subjected to ER stress. This is in contrast to previous reports where IRE1 clusters were proposed to be the primary site of XBP1 splicing; however, IRE1 clustering from XBP1s splicing has been shown to been separable previously in Ricci et al. (2019) FASEB J (where they showed that the flavinoid luteolin induces robust XBP1 splicing independent of clustering). Herein, the authors demonstrated that the clustering of IRE1-GFP is an artifact of overexpression, which is not observed upon expression of IRE1-GFP to endogenous levels. This is consistent with another recent report submitted to *eLife* from Peter Walters group showing that endogenous IRE1 does not cluster, despite previous reports (Belyy et al. 2021).

Ultimately, while the experiments appear well performed, the advance of this current manuscript is limited, although it does provide some of the controls requested of the Walter manuscript to compare to previous reports (specifically some of the experiments described in Figure 4). The data included in Figure 1-3 validate previous mechanisms proposed for XBP1 targeting to the ER using new approaches. While important to validate mechanisms using different approaches, there is no new insight included in this aspect of the work. In combination with the Walter manuscript, this work does correct the misinterpretation of the IRE1 activation mechanism resulting from overexpression artifacts, by supporting the fact that endogenous IRE1 does not appear to cluster, but instead splices XBP1 mRNA distributed through ER. Individually, this paper would not be considered strong enough to be published in *eLife*, but combined with the Walter manuscript they do correct a mechanism of IRE1 activation that is important to highlight in the literature.

*Reviewer #3 (Recommendations for the authors):*

1. Showing that the MCP-Halo and scAB-GFP do not associate with each more than chance would predict would help to at least show that the dots visualized by both techniques are not likely to be clusters of mRNAs. It is less clear to me how to show that MCP-Halo and scAB-GFP are detecting all of the relevant transcripts, but I would think this point would be important to address one way or another.

2. Would it be possible to engineer an Xbp1 RNA with both the MS2 tag and the SM tag? Presumably, in that case MCP-Halo and scAB-GFP signals should overlap for individual molecules until after splicing. More specifically for point 2, do Xbp1 RNAs that encounter IRE1 then leave the ER, or do they stay associated? Or can Xbp1 RNA leave the ER without encountering an IRE1 cluster?

3. Some statements that are in my mind not warranted:

a. Page 5, "…different from the canonical SRP-mediated recruitment…"; the SRP pathway is not examined. The pathway might use canonical SRP-mediated targeting, just relatively inefficiently.

b. Page 5, "…does not recruit XBP1 mRNA to these higher order oligomeric assemblies."; that there is no stable association is justified, but that there is no recruitment is not, in my opinion.

c. Page 12, "…drives the release of translationally active, translocon-engaged mRNAs."; there is no direct evidence for this claim, as translocons are not examined.

d. Page 12, "ER stress caused a reduction of ER association when compared to untreated conditions." There is no statistical analysis of this for these reporters, so this conclusion is not warranted.

4. Technical points:

a. The authors should explain why there is an NLS on the MCP-Halo protein.

b. The authors should state in the text or legend where the antibody used in Figure 1C recognizes XBP1 protein. If the antibody recognizes spliced wild-type XBP1 but not unspliced, that suggests that the antibody is downstream of the intron, which would also allow it to detect the HR2 mutant when unspliced, but then it shouldn't detect the HR2 mutant when spliced.

c. If the SM tag is place in frame with the spliced Xbp1 mRNA, is that species excluded from the ER membrane, as the authors' interpretations would suggest?

d. Why is 4u8C added to Figure 4F? The logic there is not clear.

[Editors’ note: further revisions were suggested prior to acceptance, as described below.]

We have considered carefully your appeal of our decision not to invite a revised version of the manuscript "Live imaging of the co-translational recruitment of XBP1 mRNA to the ER and its processing by diffuse, non-polarized IRE1α" for consideration at *eLife*. As a consequence of these deliberations, we are prepared to consider (with no guarantees of acceptance) a revised submission addressing that specific concerns that follow:

1. The authors have claimed that they "coordinated our study with the one from the Walter lab", which they provide as a reason for not needing to do certain suggested experiments/controls (since they are outlined in the paper from Belyy et al.), most notably the single molecule time-resolved tracking experiments and measurements of the IRE1 oligomeric state. Our understanding of the facts is that the papers were not submitted together as there are non adjoining cover letters indicating that these papers were linked. Thus, the papers were not considered as co-submitted. Moreover, key controls and experiments were not cross-referenced between the two papers. Only after the review process was complete for the Belyy et al. paper where we made aware of your paper. So, the current manuscript was considered as a stand alone study, and therefore we agree with the concerns of 2/3 reviewers that all of the requested experiments are imperative for publication.

If, however, revised versions of the papers are co-submitted and fully cross-referenced in the future, then in the next round of the editorial process it may be possible to consider the exclusion of some requested experiments from this study.

2. We find that in several instances your paper gives the impression that you were measuring dynamics and quantifying single molecules under these conditions, when in fact this may not have been the case. (The rebuttal letter states that the completion of the text was rushed to try to submit as soon as possible.) The authors note that the text will be edited, which is fine if, again, resubmitted papers are cross-referenced and returned to *eLife* in tandem. If the papers are returned independently without substantial cross-referencing, then the revised manuscript must stand on its own merits and dynamics and quantifying single molecules must be part of it to reach the level of novelty required by *eLife*

3. A reviewer rightfully asks for other measurements of ER stress since some splicing is apparent in the absence thapsigargin. Shorter induction times and, indeed, other measurements of ER stress should be shown.

4. We also agree that a side-by-side comparison of the tagged constructs to untagged constructs is valuable. It is more than reasonable that one should always confirm the full function of a tagged protein in relation to the untagged protein, thereby validating the behavior of the former. This would better lay out any caveats to the use of the new system, which is vital if others are to take advantage of it.

5. While the authors are indeed using a method that was previously pioneered by their lab, there are critical controls/information that should be added (e.g. spot intensity distributions) to the supplemental information.

6. The need for an experiment to measure detection efficiencies would indeed show "that the MCP-Halo and scAB-GFP do not associate with each more than chance".

7. In the rebuttal letter, it is noted that several other experiments to address the reviewers' comments are ongoing or can be started. The completion of these experiments will significantly strengthen the manuscript, and we encourage the authors to be as thorough as possible in completing these experiments.

[Editors’ note: further revisions were suggested prior to acceptance, as described below.]

Thank you for extensively editing and resubmitting your work entitled "Live imaging of the co-translational recruitment of XBP1 mRNA to the ER and its processing by diffuse, non-polarized IRE1α" for further consideration by *eLife*. As you will note below, we have asked the original three reviewers of this study to again offer their opinions and insights on your work. While they appreciate the extensive work that has gone into the revision, there are still some issues that need to be addressed prior to final acceptance of this report. Based on the reviewers' thoughtful comments, subsequent discussions, and our evaluation of the work, they are:

1. The need for improved quantitation of select data, which is required to better support some of the claims in the study (Reviewer #1).

2. Based on this analysis, a clear statement of how you are defining "puncta" in this study (Reviewer #1 and JLB). The need for this is accentuated by the fact that the companion paper also uses the word freely, and since scientists in the field will be reading both papers in tandem, it is vital that this definition is in harmony between the two studies.

3. An attempt to better position the paper, in the Abstract, Significance Statement, and Introduction, as a technological advance, rather than only a ground-breaking study on the UPR. As two of the Reviewers continue to firmly maintain (Reviewers #2-3), most of the scientific advances were embedded in the literature and/or are outlined in the companion paper.

4. With regard to this last point, please better coordinate with Belyy et al. to ensure that additional cross-references are included.

The specific comments from the Reviewers are:

*Reviewer #1 (Recommendations for the authors):*

The authors have satisfactorily addressed many of the issues raised by this reviewer. A few issues still remain unresolved.

1. Regarding the relationship between IRE1a expression level and stress inducible cluster formation, the data in figure 4 are supportive of this claim but insufficiently quantitative. Given the previously claimed importance of the IRE1a clusters in the literature, it would be important for this manuscript's claim and extremely helpful for the field for the authors to (a) provide a quantitative definition of stress inducible clusters and (b) perform quantitative analyses of the two reporter cell lines unstressed and stressed (what fraction of cells have puncta and how many puncta?). It would be even better if the authors could determine whether there is a relationship between ER intensity/IRE1a-GFP expression and propensity to form puncta, assuming only some fraction of cells form puncta, as described in the 2019 Belyy et al. PNAS paper. Note that a puncta definition would be useful for distinguishing between the structures observed in Figure 4E IRE1a-GFP and the relatively bright puncta in the "No stress" Emi1-IRE1a-GFP cell.

2. The wording of the manuscript matters. The authors have not changed their claim: "ER-poised XBP1 transcripts are processed by functional IRE1a assemblies that are homogeneously distributed throughout the ER membrane." No data are provided on what fraction of the IRE1a is in an active state.

Up to this point in the manuscript, this claim is not actually supported and the authors agree. Later, the authors cite the Belyy et al. manuscript, which also does not support the claim. There is no assay for visualizing the active form of IRE1a. The fully phosphorylated IRE1a and inactive dimer do not have physical characteristics that should significantly alter the diffusion coefficient or trajectory correlation. At this time, the Belyy et al. group can detect what appears to be dimers and tetramers/possibly larger oligomers which are presumably in the process of activating. The immunoblots do not establish what fraction of total IRE1a is phosphorylated. Therefore, the matter of where active IRE1a is distributed remains unexamined. It's an interesting question worthy of addressing. That said, the authors simply need to modify their text.

3. The authors have not mentioned sources for 4u8C, dox, tunicamycin or puromycin.

*Reviewer #2 (Recommendations for the authors):*

In the revised submission, the authors attempted to address many of the comments brought up in the previous review. However, some of the underlying problems still remain. Notably, the assay developed simply validates previous mechanistic insights into IRE1-dependent XBP1 splicing. The authors seem to agree with this in their rebuttal, making the point that this represents the first time that this process has been 'visualized' and that this assay now can be used to further probe XBP1 biology. However, this manuscript is written in such a way to suggest the focus was on improving our understanding of XBP1 splicing, not developing an assay for future work. Further, I would argue previous work clearly did a nice job of working out the mechanism, largely independent of microscopy, so I just don't see the advance here. This manuscript does provide support for a manuscript co-submitted with this revision by Peter Walter's group showing that IRE1 clusters are not required for activity, which is fine, but this is something that was previously described in published reports (see Ricci et al). I still don't feel like this work rises to the level suitable for publication in *eLife* on its own and it is relying on co-submission with the Walter manuscript to get over that bar.

*Reviewer #3 (Recommendations for the authors):*

I believe that the authors have addressed most of the technical concerns outlined, and I also find their rebuttal to be persuasive. I now appreciate that the sort of experiment I was envisioning, that underlay several of my comments (but was perhaps not obvious from them individually) would be to test the prediction that an Xbp1 mRNA would lose association with the ER membrane and its splicing could be detected by loss of scAB-GFP. But, as the authors point out, that would require simultaneous 3 channel visualization and also possibly a time scale that would be unfeasible (i.e., that the mRNA not only leaves the ER membrane), but that the ribosomes translating the GCN4 region completed their synthesis. It would also require presumably using milder conditions of ER stress, where it would be reasonable to expect some significant fraction of the Xbp1 mRNAs were unspliced.

I also think the revision benefits from more careful wording, and think the untempered claims in the original manuscript contributed to its perception.

I continue to hold the view that the majority of the paper represents a technical advance of somewhat limited general interest because this group has already established the technique, and here uses it largely to confirm a pathway that is well-accepted. I think where this manuscript complements its companion from the Walter lab is that the Walter lab shows that higher clusters of IRE1 need not form under "normal" conditions, and this paper shows that, when they do form, Xbp1 is likely not spliced there.

---

## [Author Response]

[Editors’ note: The authors appealed the original decision. What follows is the authors’ response to the first round of review.]

Comments to the Authors:While there was significant interest in the topic and preliminarily findings, we are sorry to say that, after consultation with the reviewers, we have decided that this work will not be considered further for publication by eLife.As you will read below, whilst all three reviewers concur with the importance of the paper's subject matter they independently concluded that the study would need significant additional work to become suitable for publication.

We thank the reviewers for their feedback and genuinely appreciate the time and effort spent on improving our study.

The reviewers have laid out a non-trivial number of additional controls and experiments that are considered essential. In brief, the absence of these experiments indicates that many of the claims currently lack experimental support to the level required by ELife.

We respectfully disagree. Instead, we propose that there is not a single experiment proposed by the reviewers that we could not provide in a reasonable time frame.

More specifically, the clustering analysis is of limited scope and significance,

We argue that developing a mammalian model system in which XBP1 processing clearly takes place despite the absence of visible stress-induced IRE foci is in itself a major advance that, together with the manuscript of Belyy et al. (2021), clarifies a long-standing misunderstanding about the role of IRE1 clusters.

temporal aspects of the observed events are absent, the tools used for this study need to be better validated,

Please propose experiments for further validation of the RNA imaging tools. The IRE1a-GFP construct was already validated/established by the Walter lab in 2020 (Belyy et al., 2020, PNAS). We regret not having explained this more clearly.

and single particle analysis was--in contrast to what was stated--not performed rigorously.

This statement is incorrect. Single-particle analysis was performed with state-of-the-art methodology that has also been published before (e.g. Voigt et al., 2017). We argue that there is a conceptual misunderstanding amongst the reviewers. Please see the detailed response (i.e. to reviewer 3) below.

There is also a consensus amongst the reviewers that a substantial portion of the paper is simply confirmatory based on prior work performed by the Kohno and Argon labs.

This is correct, a substantial portion of the findings are confirmatory. Which is precisely, why it serves to validate the RNA imaging tools that we have established.

At the same time our work not only confirms a model but solves a debate about how/where UPR splicing takes place, favoring a model that supports the non-polarized splicing of XBP1 and discarding the model whereby ER stress signaling would initiate from specific sites within the ER. That is, we believe, the biological importance of our work.

Reviewer #1 (Recommendations for the authors):In this study, the authors explore key steps in the mammalian unfolded protein response (UPR). They focus on the IRE1a branch, specifically, the recruitment of target mRNA XBP1 to the endoplasmic reticulum (ER) surface, the fate of the mRNA, and organization of IRE1a, itself, in response to unfolded protein stress. The novelty of this study is the use of live cell imaging techniques to follow XBP1 mRNA localization and dynamics. The translational recruitment model of XBP1 to the ER membrane surface has been previously described by others and is confirmed in live cells, here. Different reporters were used to visualize localization and translation of spliced and unspliced forms of XBP1. The authors make additional claims regarding the stress-induced clustering behavior of IRE1a. Overall, this study adds visual data, with some dynamics information to potentially help refine models of steps of XBP1 mRNA trafficking and splicing in response to ER stressors. The novelty of the study approach is undercut by the need for more thorough characterization of the reporters, better descriptions of the reagents, and more accurate descriptions of experiments and results. With the additional requested details and modifications to the text, this manuscript could be a useful addition to the ER stress literature and of broader interest to groups interested in trafficking and translation on the ER membrane.

Thank you for helping us improve our manuscript. We will be happy to provide more accurate descriptions as well as any characterization of the reporter transcripts asked for and agree that the manuscript is a useful addition to the ER stress literature.

The authors make two claims that are well supported by imaging data, mutants, and biochemical fractionation.Splicing activity by IRE1a or the presence or expression of XBP1 message in an already spliced form decreases the fraction of XBP1 associated with the ER. These observations are supported by membrane fractionation experiments.The authors have confirmed that stable ER association of unspliced XBP1 mRNA is established through HR2-dependent targeting and relies on active translation. The experimental data support this claim.Other claims are less well supported or less accurate.First, the authors claim to "directly monitor recruitment of individual XBP1 transcripts to the ER surface." I was unable to find a single movie or data series demonstrating the movement of a single particle from the cytoplasm to the ER membrane. Experiments show mRNAs on the ER or report amount of cytoplasmic vs. membrane bound. The results, presumably, of recruitment are observed and reported, not the process.

We apologize for this suboptimal phrasing and thank the reviewer for pointing it out. Of course, we are happy to provide individual example movies from amongst our large dataset to show single XBP1 transcripts as they are being recruited to the ER surface.

As the reviewer rightly points out, our analysis quantifies how wt and mutant XBP1 mRNAs are associated with the ER to different degrees. Since some of these transcripts only differ in individual nucleotide substitutions in elements involved in recruitment to the ER, we used their degree of association as a means to characterize the recruitment mechanism. At no point did we mean to insinuate that we had actually quantified recruitment dynamics.

The authors state "Next, we quantified the mobility of individual particles with respect to their ER localization and therefore assessed when an mRNA particle associates with the ER." What appears to be assessed is whether a particle is associated with the ER, not "when." I assume the authors did not mean "when" as part of temporal order, but it would be less confusing to claim to assess whether or if.

We thank the reviewer for pointing this out and did indeed not use “when” in a temporal context. We will use a different conjunction in the revised version of the manuscript.

The authors claim to show that IRE1a-catalyzed splicing mobilizes XBP1 mRNA from the ER membrane in response to ER stress. Here the data are mostly supportive of the claim. However, what the authors actually show is that splicing is necessary for an increase in XBP1 mRNA localization in the cytoplasm. No data are presented showing interactions with IRE1a and/or movement of XBP1 mRNA from the ER membrane to the cytoplasm. Based on multiple studies, the interpretation is likely correct, but the microscopy data do not show the process.

Again, we thank the reviewer for reading the manuscript attentively. What we actually show is an increase in ER association upon inhibition of IRE1a cleavage activity. Please point out how this result could be interpreted in any other way but the one proposed in the manuscript.

– We are happy to provide movies that show individual mRNAs leaving the ER, however, did not provide them in the first place because we did not consider them very informative (compared to the combined analysis of thousands of individual mRNAs and the heterogeneity of their mobility that we derived from that).

– With respect to showing interactions of individual transcripts with IRE1a clusters, we would like to point out that we did provide a specific example for this extremely rare event (Supplementary Movie 9, white arrow indicates a single XBP1 mRNA particle that colocalizes with an IRE1a cluster) to demonstrate that we were indeed able to detect those and that the absence of colocalization was no artifact of the experimental set-up.

The matter of IRE1a clustering is more problematic.The authors state, "Surprisingly, we find that XBP1 transcripts are not recruited into large IRE1a clusters, which only assemble upon overexpression of fluorescently-tagged IRE1a during ER stress. Our findings support a model where ribosome-engaged, ER-poised XBP1 mRNA is processed by functional IRE1a assemblies that are homogenously distributed throughout the ER membrane."The authors state, "ER-poised XBP1 transcripts are processed by functional IRE1a assemblies that are homogenously distributed throughout the ER membrane." No data are provided on what fraction of the IRE1a is in an active state. There is no method used to detect the active state or information on the spatial distribution of stress-activated IRE1a. This claim is not supported.

The reviewer is correct in that we do not provide any data to quantify what fraction of stress-activated IRE1a is functional or how it might be spatially distributed with respect to non-functional IRE1a assemblies. This is due to the nature of our experimental set-up and is exactly what Belyy et al. (2021) demonstrate in their complimentary manuscript. We urge the reviewer to consider both manuscripts together since they were submitted back-to-back. In respect of our agreement with Belyy et al., we refrained from performing the same single-molecule IRE1a imaging experiment that they have set-up.

However, we do demonstrate that IRE1a expressed at physiological levels is homogeneously distributed throughout the ER membrane, does not accumulate in large microscopically visible clusters (contrary to many previous reports) and that even when formed, these clusters are not sites of XBP1 processing. With this experiment, we address a major open question in the field where large IRE1a clusters have been proposed to function as the sites of XBP1 splicing for a long time.

"Our data have allowed us to visualize and uncover unanticipated features of one of the key steps of UPR initiation, the encounter of XBP1 mRNA with IRE1a to undergo splicing." This is another claim that is not quite accurate. The authors observed tagged XBP1 mRNA dynamics in co-expressing cells treated with an inhibitor of IRE1a activity. An encounter in which splicing could actually occur is never presented. Whether XBP1 mRNA is stimulated to dissociate from the ER membrane following an encounter with IRE1a-GFP is not presented. The ability of drug inhibited IRE1a-GFP to bind XBP1 mRNA is not tested or reported.

We thank the reviewer for raising this argument. As before, we should re-phrase and differentiate clearly between observing single-mRNA dynamics and the quantification of ER association of many individual mRNA transcripts, which is what we are showing in this manuscript.

The reviewer is correct, we do not show an individual mRNA molecule that is cleaved by IRE1a and leaves the ER surface in response to that. However, we do show that IRE1a is homogenously distributed throughout the ER membrane (Figure 4E) and that XBP1 transcripts are present/absent there in response to IRE1a cleavage activity (Figure 3C). This observation is not only dependent on treatment with an IRE1a inhibitor but was also observed for non-cleavable mutant transcripts that accumulate on the ER surface. We argue that the increase in ER association that we observe for these non-spliceable mutant mRNAs clearly shows that ER dissociation depends on mRNA cleavage even though we do not show (and do not claim to show) the actual dynamics of the process.

The authors have made a significant point of weighing in on the matter of IRE1a clustering in both the abstract and text. While their findings are potentially interesting, the investigation of this matter has not been sufficiently rigorous. First, the authors simply describe the attached fluorescent protein as a GFP. Is this monomeric GFP? Emerald? mNeonGreen? or? With the wide variety of fluorescent proteins available, the minimal expectation is to state what was inserted and imaged. Given the potential of many fluorescent proteins to oligomerize, the choice of fluorescent protein matters.

We apologize for the lack of specificity. As mentioned in the methods, we have reproduced the construct design established by the Walter lab in 2019. Our fusion protein design includes a GFPuv tag.

Second, the systems used may be inducing some constitutive stress. For example, in 2C, there appears to be splicing of the reporter, even in the absence of thapsigargin or expression of IRE1-fluorescent protein. Induction with Dox is for 15 hours. That's long enough for stimulation of a stress response and perhaps even some degree of adaptation. I'd like to see some other assays of ER stress before and during induction up to the time normally used to apply stress.

We thank the reviewer for this comment. Doxycycline-inducible expression of imaging reporter constructs does not induce ER stress, as indicated by the splicing rate of endogenous XBP1 mRNA or the low expression of other UPR transcripts or trigger any kind of detectable adaptation. In the experiments where splicing/translation of reporter mRNAs were characterized, we typically induced their expression for 15 hours to reach steady-state levels of mRNA expression. Please note that IRE1a-GFP expression was not controlled by Dox-inducible promoters, but by constitutive promoters.

We would be happy to include this information in the manuscript, or to repeat the characterization experiments using shorter induction times.

Similarly, in the Ire1a-KO cells in 4B, there appears to be spliced XBP1, with and without stress. Importantly, the ratio of spliced XBP1 for the emi1 variant is much lower than for wt, even though immunoblotting suggests there is significantly more EMI-1-IRE1a-GFP protein present than for wt IRE1a. Please comment. Is the GFP tagged variant less active? Is it possible that the GFP tagged variant is less activatable?

We thank the reviewer for this accurate comment. The reviewer is right, in the gel provided there is a faint band that co-migrates with the band that corresponds to the spliced mRNA. Since quantitative RT-PCR of XBP1 mRNA failed to detect any spliced XBP1 mRNA, and Western blot analysis shows no XBP1s protein in IRE1a-KO cells, this band could represent a non-specific PCR product that migrates like the “spliced” PCR band. We are conducting PCRs with a different set of primers to rule out that possibility.

Also, Emi1-IRE1a-GFP promotes splicing with lower efficiency than endogenous IRE1a, as the PCR and qPCR analyses indicate, we thank the reviewer for giving us the opportunity to mention this aspect properly.

More fundamentally, the authors claim that large IRE1a clusters… only assemble upon overexpression of fluorescently-tagged IRE1a during ER stress. At the very least, the more accurate thing would be to say fluorescently-tagged IRE1a only forms inducible observable clusters under overexpression conditions. In this study, there has been no testing of oligomeric/cluster status of native IRE1a.

We thank the reviewer for pointing this out and will re-phrase to “no observation of microscopically visible IRE1a clusters”.

There has been no use of other assays to determine if low expressed IRE1a-fluorescent protein forms oligomers of any size during stress. Nor has there been investigation of whether a version with a small epitope tag might form clusters. What about the ability of other stress conditions, such as acute thapsigargin or DTT, for the ability to induce observable clustering for the lower expressed IRE1a-GFP? Ultimately, the clustering analysis has been too limited to draw any major conclusions. The ability to cluster and its relevance are important to popular models of IRE1a activity. A nuanced study of the clustering regulation of IRE1a would be immensely useful to the field. The observation that clustering observed in mammalian cells is stress inducible, suggests that observable clustering is probably reporting on some aspect of IRE1 regulation. Clusters may represent an extreme manifestation of normal physiology. That is, even an artifact of expression does not rule out the potential importance of forming small clusters in the normal stress response. Note that the overexpressed IRE1a-GFP is constitutively maximally active. There is no stress induced increase in splicing activity, yet there is a stress-inducible change in distribution. This redistribution could reflect some sort of stress inducible regulatory step, for example sequestering of excess unengaged IRE1a, inactivated IRE1a or something else. Whatever causes visible clustering seems to be more than a simple artifact.

Thank you for raising all of these points. As above, we would like to refer to the manuscript of Belyy et al. (2021) that is also under review at *eLife* and does exactly this. The focus of our work (which purposefully does not repeat the excellent study performed in the Walter lab) is on the recruitment of XBP1 transcripts to IRE1a assemblies. We fully agree that visible clusters might have other physiologically relevant functions but simply state that they are not sites of XBP1 processing.

This is no obvious finding but has been controversially discussed in the literature for more than a decade (Li et al., 2010; Korennykh et al., 2009) and is still discussed today (Belyy et al., 2019; Tran et al., 2021) especially since HAC1 mRNAs have been shown so early on to be recruited to discrete Ire1p foci in yeast (Aragón et al., 2009; Kimata et al., 2007).

It would be helpful for the authors to compare the functionality of their constructs/system relative to untagged versions. Specifically, rates of processing. While I think the authors have demonstrated reasonably well that their XBP1 MS2 reporter does get targeted to the ER and spliced in response to ER stress, it would be useful to know if the 24 MS2 sites and attached MCS-GFP reporters (5nm+ diameter for each) affect rates of XBP1 mRNA processing. Is it possible that the construct is so large that it is relatively immobilized on the ER membrane by sterics and/or sheer size?

Thank you for proposing this experiment. While we agree with the reviewer that it is always useful to assay mRNA processing rates, our data clearly show that the ER association of our MS2 tagged reporter transcripts is abolished upon introduction of a single point mutation in the HR2 sequence (Figure 1F,G) or removal of the ER intron (Figure 3C). In addition, our RT-PCR assays clearly show that reporter transcripts are efficiently spliced only in response to ER stress (Figure 1B,C).

Therefore, we do not understand how differences in processing rates (compared to endogenous transcripts) would affect the findings presented here, where the same kind of bias (potentially introduced by the MS2 labeling) affects all reporter transcripts in the same way. The observed differences in ER association in between different reporters can therefore only be due to differences in experimental set-up (mutants, inhibitors) and are not caused by the labeling method.

Could the size of the reporter slow the approach of IRE1a to the cleavage site? Does high expression of the XBP1 substrate impact the efficiency of processing? That is, if client is saturating, then encountering and splicing by activated IRE1a on the ER membrane could compensate for potential issues due to size or mobility.

All of these are fair questions and could be assessed at a later time. However, they are not relevant to our main conclusions, which are qualitative (microscopically visible IRE1a clusters are not sites of XBP1 processing) and highly relevant to the research community.

In addition, it would be useful to determine how much XBP1 reporter mRNA is expressed and how this compares to endogenous levels of XBP1 mRNA in unstressed and stressed cells.

We can provide this information based on the qPCR assays that we have performed.

Reviewer #2 (Recommendations for the authors):This manuscript develops a different reporters to monitor XBP1 targeting to the ER, which are used to confirm previous results showing that XBP1 is directed to the ER through a mechanism involving translation of the HR2 mRNA sequence. As indicated in the manuscript, this mechanism had been previously reported by Kohno, and, while the work presented here confirms this model, it does not extend it.

Please see comment above. The experiments provide important validation and were used to establish the single molecule imaging tools, which will be extremely useful for further characterization of mRNA processing during the UPR on the ER.

The major advance from this manuscript, apart from the reporter development, relates to the fact that IRE1 clusters are not observed in cells expressing endogenous levels of IRE1-GFP and subjected to ER stress. This is in contrast to previous reports where IRE1 clusters were proposed to be the primary site of XBP1 splicing; however, IRE1 clustering from XBP1s splicing has been shown to been separable previously in Ricci et al. (2019) FASEB J (where they showed that the flavinoid luteolin induces robust XBP1 splicing independent of clustering). Herein, the authors demonstrated that the clustering of IRE1-GFP is an artifact of overexpression, which is not observed upon expression of IRE1-GFP to endogenous levels. This is consistent with another recent report submitted to ELIFE from Peter Walters group showing that endogenous IRE1 does not cluster, despite previous reports (Belyy et al. 2021).

Please see comment above. Apart from the study of Ricci and colleagues that the reviewer mentions, earlier works from the Walter lab (Belyy et al. PNAS 2020) as well as their manuscript currently under review at *eLife* (Belyy et al., 2021), are consistent with our observation that large IRE1a foci only form whenever IRE1a is overexpressed.

More importantly, however, we show for the first time that the ability of such foci to recruit IRE1a is indeed very limited. The direct observation of this lack of recruitment of XBP1 mRNA solves a long-standing debate in the field. Of course, our findings do not rule out the possibility that under specific conditions or in specific cell types (for instance, in multiple myeloma cells, where IRE1a is strongly overexpressed), these foci may be able to recruit other RNA substrates, or to promote other processes in the stressed cell.

Ultimately, while the experiments appear well performed, the advance of this current manuscript is limited, although it does provide some of the controls requested of the Walter manuscript to compare to previous reports (specifically some of the experiments described in Figure 4). The data included in Figure 1-3 validate previous mechanisms proposed for XBP1 targeting to the ER using new approaches. While important to validate mechanisms using different approaches, there is no new insight included in this aspect of the work.

We agree with the reviewer but would like to stress that even if XBP1 targeting mechanisms have been proposed and discussed in the literature for a while, our study is the first to directly test them.

In combination with the Walter manuscript, this work does correct the misinterpretation of the IRE1 activation mechanism resulting from overexpression artifacts, by supporting the fact that endogenous IRE1 does not appear to cluster, but instead splices XBP1 mRNA distributed through ER. Individually, this paper would not be considered strong enough to be published in eLife, but combined with the Walter manuscript they do correct a mechanism of IRE1 activation that is important to highlight in the literature.

We thank the reviewer for this important comment.

Reviewer #3 (Recommendations for the authors):1. Showing that the MCP-Halo and scAB-GFP do not associate with each more than chance would predict would help to at least show that the dots visualized by both techniques are not likely to be clusters of mRNAs. It is less clear to me how to show that MCP-Halo and scAB-GFP are detecting all of the relevant transcripts, but I would think this point would be important to address one way or another.

To answer your question concerning detection efficiencies: This could be done in e.g. a FISH experiment where two different probe sets are used to detect the same transcript species. The degree of colocalization between both labels can then be used to calculate detection efficiencies for each fluorescent label. For further information please see Voigt*, Gerbracht* et al., 2019, Nat Prot. We are happy to provide such an experiment in a revised manuscript if needed.

2. Would it be possible to engineer an Xbp1 RNA with both the MS2 tag and the SM tag? Presumably, in that case MCP-Halo and scAB-GFP signals should overlap for individual molecules until after splicing. More specifically for point 2, do Xbp1 RNAs that encounter IRE1 then leave the ER, or do they stay associated? Or can Xbp1 RNA leave the ER without encountering an IRE1 cluster?

This is precisely what we have done. Please see the cartoon in Figure 2A. XBP1u translation reporter transcripts contain both, the SM and MS2 tags. This is why MCP-Halo and scAB-GFP spots need to colocalize. XBP1 mRNAs that are processed by IRE1a eventually leave the ER as illustrated by differences in ER association upon induction of ER stress and treatment with 4µ8C in Figure 3C. The dynamics of the cleavage reaction cannot be inferred from the kind of live imaging experiment that we have performed here.

Last, yes, it is possible that XBP1 mRNAs could leave the ER without encountering IRE1a, yet, this would only mean that we are underestimating the efficiency of XBP1 recruitment to the ER. Since our conclusions from the ER association analysis are qualitative, not quantitative, we believe that this possibility is of minor relevance.

3. Some statements that are in my mind not warranted:a. Page 5, "…different from the canonical SRP-mediated recruitment…"; the SRP pathway is not examined. The pathway might use canonical SRP-mediated targeting, just relatively inefficiently.

Please excuse the vague phrasing. This is exactly what we mean. As well as the lack of a canonical signal sequence in the XBP1 ORF.

b. Page 5, "…does not recruit XBP1 mRNA to these higher order oligomeric assemblies."; that there is no stable association is justified, but that there is no recruitment is not, in my opinion.

We will re-phrase this sentence.

c. Page 12, "…drives the release of translationally active, translocon-engaged mRNAs."; there is no direct evidence for this claim, as translocons are not examined.

This is correct. We are working on this at the moment.

d. Page 12, "ER stress caused a reduction of ER association when compared to untreated conditions." There is no statistical analysis of this for these reporters, so this conclusion is not warranted.

This is correct. We will add the analysis.

4. Technical points:a. The authors should explain why there is an NLS on the MCP-Halo protein.

To recruit excess/unbound MCP-Halo protein away from the cytoplasm and thereby increase signal/noise. We will add this detail to the revised manuscript.

b. The authors should state in the text or legend where the antibody used in Figure 1C recognizes XBP1 protein. If the antibody recognizes spliced wild-type XBP1 but not unspliced, that suggests that the antibody is downstream of the intron, which would also allow it to detect the HR2 mutant when unspliced, but then it shouldn't detect the HR2 mutant when spliced.

As indicated by the provider (Santa Cruz Biotechnology), the antibody recognizes a region within the N-terminal half of XBP1 protein, encoded upstream the UPR intron. Thus, this antibody should recognize both XBP1u and XBP1s proteins. Yet, we can only detect XBP1s protein. We believe this lack of recognition is due to the translational regulation of XBP1u mRNA and the low stability of XBP1u protein.

c. If the SM tag is place in frame with the spliced Xbp1 mRNA, is that species excluded from the ER membrane, as the authors' interpretations would suggest?

This experiment is in progress.

d. Why is 4u8C added to Figure 4F? The logic there is not clear.

In order to inhibit IRE1a cleavage activity and allow accumulation/detection of XBP1 mRNA in IRE1a clusters. We have performed the experiment with the same result in absence and presence of the compound. Here, we show the lack of accumulation in 4µ8C treated cells to illustrate that even a “kiss-and-run” mechanism (very fast turnover) could be detected in this experimental set-up.

[Editors’ note: what follows is the authors’ response to the second round of review.]

1. The authors have claimed that they "coordinated our study with the one from the Walter lab", which they provide as an reason for not needing to do certain suggested experiments/controls (since they are outlined in the paper from Belyy et al.), most notably the single molecule time-resolved tracking experiments and measurements of the IRE1 oligomeric state. Our understanding of the facts is that the papers were not submitted together as there are non adjoining cover letters indicating that these papers were linked. Thus, the papers were not considered as co-submitted. Moreover, key controls and experiments were not cross-referenced between the two papers. Only after the review process was complete for the Belyy et al. paper where we made aware of your paper. So, the current manuscript was considered as a stand alone study, and therefore we agree with the concerns of 2/3 reviewers that all of the requested experiments are imperative for publication.

Thank you for explaining the details of the editorial process, through which our manuscript proceeded. While we did indeed submit our manuscript a few weeks after the Belyy et al. paper, both cover letters actually mentioned the other’s manuscript and suggested a parallel revision process. We regret that this could not happen in a more coordinated manner and aimed to address both, missing controls as well as cross-references, in the revised version of the manuscript.

If, however, revised versions of the papers are co-submitted and fully cross-referenced in the future, then in the next round of the editorial process it may be possible to consider the exclusion of some requested experiments from this study.

We co-submit our revised and cross-referenced manuscript together with the revised manuscript of Belyy et al..

2. We find that in several instances your paper gives the impression that you were measuring dynamics and quantifying single molecules under these conditions, when in fact this may not have been the case. (The rebuttal letter states that the completion of the text was rushed to try to submit as soon as possible.) The authors note that the text will be edited, which is fine if, again, resubmitted papers are cross-referenced and returned to eLife in tandem. If the papers are returned independently without substantial cross-referencing, then the revised manuscript must stand on its own merits and dynamics and quantifying single molecules must be part of it to reach the level of novelty required by eLife.

We apologize for any confusion caused by the single-molecule terminology used.

Our study indeed quantifies single particles and their degree of association with the ER using both live and fixed single-molecule imaging approaches. While it is correct that we quantify instantaneous diffusion coefficients as a measure of the mobility (or “dynamics”) of individual mRNA particles, this does not mean that we are also able to characterize the “recruitment dynamics” (as in the complete trajectory travelled) of individual mRNAs on their way to the ER. This is due to experimental limitations such as the high particle number and mobility as well as low signal/noise and rapid photo bleaching of diffraction-limited spots (as which we detect single particles) that are inherent to any single-molecule imaging experiment. Instead, our analysis relies on the quantification of the mobility and subcellular localization of single particles that were imaged over short periods of time at high temporal resolution.

In summary, our work relies on the quantification of single molecules and their diffusive properties (“dynamics”) over short time frames (ms scale). However, this is not the same as the analysis of “recruitment dynamics” that would involve tracking individual mRNAs over the extend of time (min scale) that a particle requires to travel after translation initiation in the cytoplasm to the ER surface, which is not possible in currently available live single particle imaging set-ups.

We thank the reviewers for pointing out how confusingly we have applied the term “dynamics” and have made sure to re-phrase accordingly in the revised version of our manuscript.

3. A reviewer rightfully asks for other measurements of ER stress since some splicing is apparent in the absence thapsigargin. Shorter induction times and, indeed, other measurements of ER stress should be shown.

We thank the reviewers for requesting a more thorough analysis of the behavior of our reporter mRNAs and their effect on ER stress signaling. We now include a more detailed characterization of UPR signaling in the cell lines used in this study both under non-stress and stress conditions (Figure 1—figure supplement 1 and Figure 4—figure supplement 1).

4. We also agree that a side-by-side comparison of the tagged constructs to untagged constructs is valuable. It is more than reasonable that one should always confirm the full function of a tagged protein in relation to the untagged protein, thereby validating the behavior of the former. This would better lay out any caveats to the use of the new system, which is vital if others are to take advantage of it.

We agree with the reviewer’s sensible comment and include these control experiments (Figure 4—figure supplement 1) in the updated version of our manuscript.

5. While the authors are indeed using a method that was previously pioneered by their lab, there are critical controls/information that should be added (e.g. spot intensity distributions) to the supplemental information.

Thank you for pointing this out. We now provide intensity distributions for single-particle signal from live (Figure 3figure supplement 1E) as well as fixed cell (Figure 2—figure supplement 1D) imaging experiments.

6. The need for an experiment to measure detection efficiencies would indeed show "that the MCP-Halo and scAB-GFP do not associate with each more than chance".

To address this concern and test if MCP-Halo and scAB-GFP could associate unspecifically, we have performed a targeted imaging experiment (Figure 2—figure supplement 2A), for which we provide data (Figure 2—figure supplement 2B) and quantification (Figure 2—figure supplement 2C) in the revised manuscript.

In brief, we imaged mRNA reporters that encode MCP-Halo and scAB-GFP binding sites either combined on a single or separately on two individual transcripts. Image data analysis shows that the fraction of co-localizing MCP-Halo and scAB-GFP spots is reduced to background levels (Mean ± SD = 0.01 ±0.03) when binding sites are co-expressed from separate reporter transcripts in the same cell (as opposed to 0.50 ±0.17 when expressed from a single transcript), and thus demonstrates that scAB-GFP and MCP-Halo do not associate or co-localize by more than chance (Figure 2—figure supplement 2C).

7. In the rebuttal letter, it is noted that several other experiments to address the reviewers' comments are ongoing or can be started. The completion of these experiments will significantly strengthen the manuscript, and we encourage the authors to be as thorough as possible in completing these experiments.

Several experiments to characterize the translation of XBP1 reporters on the ER are on their way. We have performed ribosome run-off experiments (after Harringtonine treatment) to characterize translation elongation rates on ER associated transcripts and found that ribosome occupancy was too low to fit our data and quantify specific elongation rates. Thus, we are now in the process of engineering a new, brighter XBP1u translation site reporter. In addition, we have also started to generate XBP1s reporter cell lines. Unfortunately, their generation was beyond the time frame that we agreed on for the coordinated re-submission of the two manuscripts.

[Editors’ note: what follows is the authors’ response to the second round of review.]

1. The need for improved quantitation of select data, which is required to better support some of the claims in the study (Reviewer #1).

We have added the quantification in Figure 4F.

2. Based on this analysis, a clear statement of how you are defining "puncta" in this study (Reviewer #1 and JLB). The need for this is accentuated by the fact that the companion paper also uses the word freely, and since scientists in the field will be reading both papers in tandem, it is vital that this definition is in harmony between the two studies.

Thank you for pointing this out. We now define IRE1a-GFP foci as intensity aggregates that can be detected as an enrichment of GFP intensity, which is ≥ 5-fold over background. Similarly, we define cells as containing IRE1a-GFP foci if at least 1% of all cellular GFP signal is contained in foci. This quantification is in accordance with the quantification employed in the manuscript of Belyy et al.

3. An attempt to better position the paper, in the Abstract, Significance Statement, and Introduction, as a technological advance, rather than only a ground-breaking study on the UPR. As two of the Reviewers continue to firmly maintain (Reviewers #2-3), most of the scientific advances were embedded in the literature and/or are outlined in the companion paper.

We have modified Abstract, Significance Statement and Introduction accordingly.

4. With regard to this last point, please better coordinate with Belyy et al. to ensure that additional cross-references are included.

We have included additional cross references and discussed this with Belyy et al.

The specific comments from the Reviewers are:Reviewer #1 (Recommendations for the authors):The authors have satisfactorily addressed many of the issues raised by this reviewer. A few issues still remain unresolved.1. Regarding the relationship between IRE1a expression level and stress inducible cluster formation, the data in figure 4 are supportive of this claim but insufficiently quantitative. Given the previously claimed importance of the IRE1a clusters in the literature, it would be important for this manuscript's claim and extremely helpful for the field for the authors to (a) provide a quantitative definition of stress inducible clusters and (b) perform quantitative analyses of the two reporter cell lines unstressed and stressed (what fraction of cells have puncta and how many puncta?). It would be even better if the authors could determine whether there is a relationship between ER intensity/IRE1a-GFP expression and propensity to form puncta, assuming only some fraction of cells form puncta, as described in the 2019 Belyy et al. PNAS paper. Note that a puncta definition would be useful for distinguishing between the structures observed in Figure 4E IRE1a-GFP and the relatively bright puncta in the "No stress" Emi1-IRE1a-GFP cell.

Thank you for this input. We have added a quantification of our data in Figure 4F. In agreement with Belyy et al., we define cells as IRE1-foci containing if ≥ 1% of the total cellular IRE1a-GFP fluorescence can be attributed to puncta that are detected via thresholding (of normalized images) using a 5-fold fluorescence intensity enrichment over background as cut-off.

2. The wording of the manuscript matters. The authors have not changed their claim: "ER-poised XBP1 transcripts are processed by functional IRE1a assemblies that are homogeneously distributed throughout the ER membrane." No data are provided on what fraction of the IRE1a is in an active state.Up to this point in the manuscript, this claim is not actually supported and the authors agree. Later, the authors cite the Belyy et al. manuscript, which also does not support the claim. There is no assay for visualizing the active form of IRE1a. The fully phosphorylated IRE1a and inactive dimer do not have physical characteristics that should significantly alter the diffusion coefficient or trajectory correlation. At this time, the Belyy et al. group can detect what appears to be dimers and tetramers/possibly larger oligomers which are presumably in the process of activating. The immunoblots do not establish what fraction of total IRE1a is phosphorylated. Therefore, the matter of where active IRE1a is distributed remains unexamined. It's an interesting question worthy of addressing. That said, the authors simply need to modify their text.

Thank you for pointing this out. We had originally misunderstood the reviewer’s point and have now modified the text to reflect that we have no information on what fraction of IRE1a assemblies is functional or how it is distributed throughout the membrane.

3. The authors have not mentioned sources for 4u8C, dox, tunicamycin or puromycin.

We have added their sources in the methods section.

Reviewer #2 (Recommendations for the authors):In the revised submission, the authors attempted to address many of the comments brought up in the previous review. However, some of the underlying problems still remain. Notably, the assay developed simply validates previous mechanistic insights into IRE1-dependent XBP1 splicing. The authors seem to agree with this in their rebuttal, making the point that this represents the first time that this process has been 'visualized' and that this assay now can be used to further probe XBP1 biology. However, this manuscript is written in such a way to suggest the focus was on improving our understanding of XBP1 splicing, not developing an assay for future work. Further, I would argue previous work clearly did a nice job of working out the mechanism, largely independent of microscopy, so I just don't see the advance here. This manuscript does provide support for a manuscript co-submitted with this revision by Peter Walter's group showing that IRE1 clusters are not required for activity, which is fine, but this is something that was previously described in published reports (see Ricci et al). I still don't feel like this work rises to the level suitable for publication in eLife on its own and it is relying on co-submission with the Walter manuscript to get over that bar.

Seeing is believing, which is why we strongly advocate the development of single-molecule tools that can test biological hypotheses. In addition, we have modified Abstract, Significance Statement and Introduction to also highlight the methodological advance of our work.